# Beyond Data Scarcity: A Frequency-Driven Framework for Zero-Shot Forecasting

## Abstract

Time series forecasting is critical in numerous real-world applications, requiring accurate predictions of future values based on observed patterns. While traditional forecasting techniques work well in in-domain scenarios with ample data, they struggle when data is scarce or not available at all, motivating the emergence of zero-shot and few-shot learning settings. Recent advancements often leverage large-scale foundation models for such tasks, but these methods require extensive data and compute resources, and their performance may be hindered by ineffective learning from the available training set. This raises a fundamental question: *What factors influence effective learning from data in time series forecasting?* Toward addressing this, we propose using Fourier analysis to investigate how models learn from synthetic and real-world time series data. Our findings reveal that forecasters commonly suffer from poor learning from data with multiple frequencies and poor generalization to unseen frequencies, which impedes their predictive performance. To alleviate these issues, we present a novel synthetic data generation framework, designed to enhance real data or replace it completely by creating task-specific frequency information, requiring only the sampling rate of the target data. Our approach, *Freq-Synth*, improves the robustness of both foundation as well as non-foundation forecast models in zero-shot and few-shot settings, facilitating more reliable time series forecasting under limited data scenarios.

## 1 Introduction

Time series forecasting (TSF) plays a critical role in various areas, such as finance, healthcare, and energy, where accurate predictions of future values are essential for decision-making and planning. Traditionally, in-domain learning has been the common setting for developing forecasting models, where a model is trained using data from the same domain it will later be deployed in (Salinas et al., 2020; Zhou et al., 2021). This ensures that the model captures the patterns, seasonality, and trends specific to the target domain, improving its predictive performance. However, a significant challenge arises when there is scarce or no historical information available for training, limiting the ability to apply traditional in-domain learning approaches (Sarmas et al., 2022; Fong et al., 2020). In such cases, the emergence of zero-shot (ZS) and few-shot (FS) learning settings offer potential solutions. Zero-shot learning enables models to generalize to new, unseen domains without requiring domain-specific data by leveraging knowledge transfer from other domains or tasks. Few-shot learning, on the other hand, allows fine-tuning on limited amounts of domain-specific data. In this paper, we focus mostly on the FS and ZS TSF setting, considering limited train data or its complete absence.

Zero-shot and few-shot techniques for TSF are often built upon foundation models, which are pre-trained on vast amounts of diverse data and can generalize to a wide range of tasks (Das et al., 2024; Ansari et al., 2024). However, foundation models (FMs) face several challenges, such as their huge data requirements, high computational costs, difficulties in fine-tuning for specific applications, and the risk of model over-generalization, which can lead to sub-optimal performance on specialized tasks (Ekambaram et al., 2024). Moreover, foundation models often struggle to fully exploit the train distribution, limiting their ability to capture domain-specific patterns crucial for accurate zero-shot time series forecasting (see Sec. 4). A potential approach to alleviate data and compute limitations is to train models, including non-foundational, on task-specific *synthetic* information (Dooley et al., 2024), thus eliminating real-data requirements and reducing compute time. Unfortunately, the factors that govern effective learning from synthetic data using non-foundation models remain unclear, and our work aims to advance the general understanding of this challenge.

We advocate throughout this paper that *Fourier analysis* is the natural framework for assessing the effectiveness of synthetic data in TSF (Yi et al., 2023). Fourier analysis decomposes a signal into its constituent frequencies, allowing for a detailed examination of how different components contribute to the overall structure of the data. Its key advantages include the ability to reveal periodic patterns, smooth out noise, and identify important frequency-based features that might not be apparent in the temporal or spatial domain (Körner, 2022). Under this lens, issues of overfitting and underfitting can be understood as forms of *frequency generalization* and *frequency confusion*, which describe the ability to generalize to unseen periodic patterns or struggle with the inference of in-domain periods, respectively (see Sec. 4). Further, by analyzing synthetic data through Fourier transforms, one can more clearly visualize how well the data captures the true underlying patterns of the target domain. This ultimately leads to a straightforward and structured procedure for generating synthetic data that alleviates confusion and improves generalization, avoiding over-representation of irrelevant details while preserving key structural components.

By harnessing Fourier analysis in the context of time series forecasting using non-foundational and foundational models, we illustrate several shortcomings of such techniques. First, we observe that increasing the available frequencies of the train set while fixing those of the test set leads to inferior test results. Second, we find the test performance to be positively related to the alignment of the train and test sets in the frequency space. Finally, we demonstrate that foundation models overfit to certain frequencies, thus under-performing on general frequencies. Based on our findings, we propose a straightforward observation to generating synthetic data: *the train set should span the predominant frequencies of the target domain*. While intuitive, our observation is often infeasible to code, as the span of target frequencies is unknown. Instead, we design a simple heuristic, allowing to generate lightweight yet target-oriented synthetic data, given the sampling rate of the target domain. We evaluate our approach in the zero-shot and few-shot settings on recent state-of-the-art models, trained on real vs. synthetic data. Our results highlight the effectiveness of our approach, *Freq-Synth*, and its advantages in comparison to other methods. Our main contributions include:

1. We analyze the importance of frequencies in time series models, especially in the context of transfer learning. We introduce the concepts frequency confusion and frequency generalization, which facilitate the identification of potential challenges in ZS forecasting.
2. Given the target sampling rate, we propose a simple, easy-to-code, and efficient time series synthetic generator whose data is small in scale yet effective for FS and ZS tasks.
3. Through extensive evaluations, we show that our synthetic data obtains improved ZS and even FS measures on several non-foundation and foundation models across several tasks.

## 2 RELATED WORK

**In-domain time series forecasting.**   For decades, non-deep statistical TSF models held the state-of-the-art (SOTA) status (Makridakis & Hibon, 2000; Makridakis et al., 2008), but in recent years, purely neural network-based SOTA approaches for TSF have emerged (Salinas et al., 2020; Oreshkin et al., 2020). Rapid development has led to various techniques including the usage of trend and seasonality (Zhou et al., 2022), patching time series (Nie et al., 2023), exploiting inter-channel relations (Liu et al., 2024b), and many others (Wu et al., 2021; Zhang & Yan, 2023; Xu et al., 2024). These approaches, however, have been considered in the in-domain setting, where there is an available train set that statistically aligns with the test set.

**Zero-shot and few-shot TSF.**   While several attempts have been made in utilizing non-foundation models for ZS and FS TSF (Orozco & Roberts, 2020; Oreshkin et al., 2021; Jin et al., 2022), interest has quickly shifted to foundation models. Specifically Large Language Models (LLMs) are commonly considered, using various backbones including GPT-2 (Zhou et al., 2023; Liu et al., 2024a), LLaMA (Jin et al., 2024), and others (Gruver et al., 2024). Additional methods exploit trend-seasonality-residual decompositions (Cao et al., 2024) and Transformer blocks (Goswami et al., 2024). One of the main limitations of foundation models is their dependence on large volumes of data. Thus, recent studies have incorporated synthetic information alongside real data involving seasonal patterns and trends (Das et al., 2024) and Gaussian processes (Ansari et al., 2024). Closely related to our work is ForecastPFN (Dooley et al., 2024), where the authors perform zero-shot time series forecasting by training solely on synthetic data. Still, we argue that the factors determining effective learning in such settings remain unclear, particularly for non-foundation models.

**Fourier analysis in time series applications.** Fourier analysis and spectral theory are commonly used in various modern machine learning tasks (Yi et al., 2023). Incorporating frequency information has been done via compression (Rippel et al., 2015), data augmentation (Yang & Hong, 2022), and neural operators (Li et al., 2021). Examples in neural network design use real-valued (Xu et al., 2020) and complex-valued (Cao et al., 2020) representations. Other works span across anomaly detection (Ren et al., 2019), classification (Wang et al., 2018; Zhang et al., 2022), and time series forecasting (Zhang et al., 2017; Woo et al., 2022). Recent works have also introduced model design modification to support periodic pattern embedding (Ekambaram et al., 2024; Liu et al., 2024a; Cao et al., 2024), some of which are employed in work. Closely related to our work are studies that considered synthetic data including many frequencies (Das et al., 2024; Ansari et al., 2024; Dooley et al., 2024). In this paper, we further extend this research direction and harness Fourier analysis to study synthetic data and its effect on zero-shot time series forecasting.

## 3 BACKGROUND

To motivate our analysis and approach to zero-shot and few-shot forecasting as discussed in Sec. 4, we present basic concepts and results related to Fourier analysis and time series information, see also (Shumway & Stoffer, 2000). It is well-known that for any time series sample $x_1, \ldots, x_n \subset \mathbb{R}$ and under carefully chosen coefficients, we have for odd $n$ that

$$x_t = a_0 + \sum_{j=1}^{(n-1)/2} \left[ a_j \cos(2\pi t\, j/n) + b_j \sin(2\pi t\, j/n) \right] , \tag{1}$$

for $t = 1, \ldots, n$, $a_0$ is the bias, $a_j$ and $b_j$ are the amplitude coefficients, and $t \in \mathbb{Z}$. The frequencies $\omega_j := j/n$ represent cycles per time unit, where a cycle is a complete period of the cosine or sine, e.g., for $\omega = 0.5$, the series makes two cycles per time unit. We also consider an equivalent form, obtained via a trigonometric identity of Eq. 1 and given by

$$x_t = a_0 + \sum_{j=1}^{(n-1)/2} A_j \cos(2\pi t\, \omega_j + \phi_j) , \tag{2}$$

where the amplitude $A_j = \sqrt{a_j^2 + b_j^2}$ and $\phi_j = \tan^{-1}(b_j/a_j)$ is the phase of the $j$th frequency, express the standard deviation and the cosine function starting point respectively. Notably, dominant periodic components in a signal are associated with larger amplitudes.

Another important concept for our work is the *periodogram* (Schuster, 1898). We define the scaled periodogram, which is closely related to the amplitude $A_j$, and it is defined via

$$P(\omega_j) = A_j^2 , \tag{3}$$

where large values of $P(\omega_j)$ correspond to predominant *fundamental frequencies* $j/n$, whereas small values of $P(\omega_j)$ can be viewed as noise. In practice, the scaled periodogram can be estimated via the discrete Fourier transform (DFT), which represents a weighted average of the data $d(\omega_j) = n^{-1/2} \sum_{t=1}^{n} x_t \exp(-2\pi i t\, j/n)$, with $i$ the imaginary number. It follows that $P(\omega_j) = \frac{4}{n}|d(\omega_j)|^2$. Finally, *Harmonics* represent frequencies of the form $k\bar{\omega}_j$ for a dominant fundamental frequency $\bar{\omega}_j$, $k \in \mathbb{N}$. They appear in time series data when non-sinusoidal components arise, and contribute to the structure of the signal. In what follows, we will show that harmonics, as depicted in the periodogram, are crucial in understanding the effect of data on zero-shot and few-shot learning and information transfer in large time series models.

## 4 FOURIER ANALYSIS AND GENERATION FOR ZERO-SHOT TSF

Many existing approaches for zero-shot TSF are based on large foundation models (Ansari et al., 2024). These neural networks are computationally demanding and need large volumes of data for training. In this work, we aim to maximize the learning efficiency from data, with the goal of reducing data and compute requirements, especially in the ZS and FS settings, where data is scarce or unavailable. Particularly, we are interested in answering the following overarching question:

*What factors govern effective learning in zero-shot time series forecasting?*

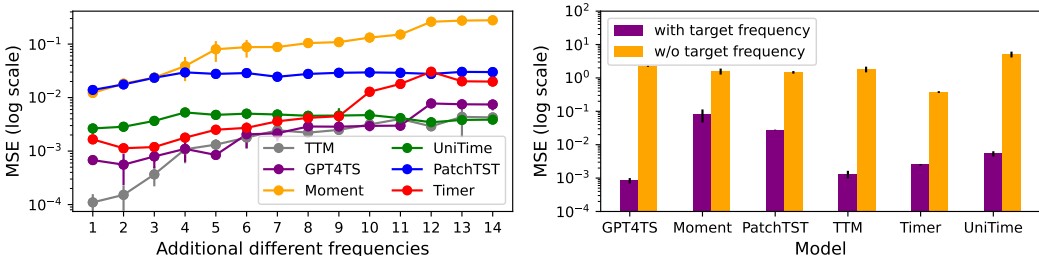

Figure 1: We show an example of frequency confusion, where adding more frequencies gradually degrades performance (left). We also observe large performance differences when the fundamental target frequency exist vs. absent in the train set, implying poor frequency generalization (right).

Understanding such factors better may lead to reducing data requirements by generating compact task-specific synthetic data. Similarly, compute reduction can be achieved by using non-foundation models on that data. Ultimately, if we succeed in answering the above question, we could potentially employ *non-foundation* models for solving zero-shot TSF, training solely on *synthetic data*.

### 4.1 FREQUENCY CONFUSION AND FREQUENCY GENERALIZATION

Toward uncovering the factors that determine effective learning, we examine time series forecasting through the lens of Fourier analysis. Specifically, to quantify the differences between the train and test sets and their corresponding forecasting errors, we will use the periodogram (see Sec. 3) and the following two new frequency-based concepts.

**Definition 4.1** (Frequency Confusion). A performance degradation observed in the case where the train set consists of the target frequencies along with other, unrelated, frequencies.

**Definition 4.2** (Frequency Generalization). The model's ability to perform well during inference on data with frequencies that were unavailable during training.

In other words, Def. 4.1 describes the model's difficulty in learning from multiple frequencies, where some may be unnecessary. It is closely related to *capacity*, introduced in (Han et al., 2024) to assess data fit, and *domain confusion* (Liu et al., 2024a) related to datasets from different domains. Def. 4.2, on the other hand, deals with the ability of a trained model to obtain consistent performance across learned as well as unseen frequencies. This definition is closely related to *domain generalization* (Wang et al., 2022), where there, the generalization is in the context of performing well on datasets of different domains.

Equipped with these definitions, we consider experiments, aiming to identify whether frequency confusion and frequency generalization assist in understanding model behavior. For these experiments, we use recent non-foundation and foundation state-of-the-art (SOTA) TSF models including GPT4TS (Zhou et al., 2023), Moment (Goswami et al., 2024), PatchTST (Nie et al., 2023), TTM (Ekambaram et al., 2024), Timer (Liu et al., 2024d), and UniTime (Liu et al., 2024a). The first experiment trains the above models on a simple sine wave dataset with $\omega = 1/24$, representing an hour to daily based sampling rate. From here and throughout our discussion, we interchangeably use the terms sampling rate and frequency. Then, we incrementally add additional sine waves with various frequencies to the train set, re-train, and measure the prediction error of the sine wave. We plot in Fig. 1 (left) the mean squared error (MSE) of the prediction in log scale vs. the number of additional train frequencies. Importantly, in all cases the model has access to the original data, and thus to the fundamental target sampling rate. Notably, all models present an increase in test MSE, even if mild, as more frequencies are added, suggesting that they suffer from frequency confusion. In the second experiment, we use the same dataset and models. However, now every model is trained twice: on a train set including the target sampling rate (frequency), and on a train set without it. As before, we plot the test MSE errors in log scale and present the results in Fig. 1 (right). The bar chart shows that in all cases, having access to the target frequency (purple) leads to significant performance gains in comparison to training without that frequency (orange). This experiment suggests that deep TSF models may struggle to generalize to unseen frequencies, even on simple toy examples.

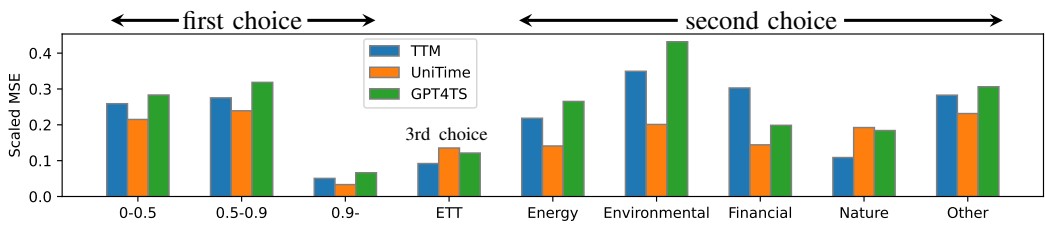

Figure 2: Transfer learning performance bars for various frequency-based alignments between the train and test sets. Ranged values represent the periodogram PCC (left), ETT is a single dataset with different sampling rates (middle), and the remaining are sector based categories (right).

The third experiment explores the effect of similar and dissimilar frequency spaces in the context of transfer learning scenarios. Here, we use GPT4TS, TTM, and UniTime, and we utilize a pool of datasets, see Fig. 10. We train each model separately on every dataset in the pool, and we use the learned model to infer over all the remaining datasets. To make the test MSE of different datasets comparable, we perform min-max normalization per dataset (scaled MSE). We organize the results into clusters based on the following choices: 1) The train and test sets share similar or dissimilar frequencies, as measured by the periodogram. 2) The train and test sets are from similar domains or sectors, e.g., energy-related information. 3) The train and test sets are sampled from the same dataset (ETT), but with different sampling rates (frequencies). Specifically, ETTh1 and ETTm1 are sampled at an hour rate and per 15 minutes, respectively. We present in Fig. 2 the scaled MSE measures of this experiment with respect to the above mentioned choices. Particularly, the three leftmost bar groups correspond to the Pearson correlation coefficient (PCC) of the periodogram between datasets (Choice 1). Then, the five rightmost bar groups are various sector domains (Choice 2). Finally, ETT is the electricity transformer temperature dataset (Choice 3). The results show that while same sector training may help (e.g., TTM on nature), the general performance is inconsistent. Similarly, using the same data (ETT) lowers the scaled MSE, however, it is still higher than training on datasets whose PCC is highly correlated in the frequency space (PCC $\geq 0.9$).

The above analysis reinforces the emergence of Fourier analysis as a key tool for determining the factors that affect effective learning. Moreover, it leads to the following straightforward observation: *training on datasets that share a similar periodogram with the target data improves the results of deep neural networks for TSF, in zero-shot scenarios and more generally.* Unfortunately, the latter observation is infeasible to code in practice, as we do not know the full target frequency distribution in zero-shot tasks. How can this observation used in practice? We propose in the next subsection a simple heuristic for generating synthetic information that we found to be highly effective.

### 4.2 FREQ-SYNTH: SYNTHETIC TIME SERIES BASED ON FUNDAMENTAL FREQUENCIES

Following our analysis, we propose a simple yet effective approach to zero-shot TSF, which we refer to as *Freq-Synth*. Namely, we generate task-specific synthetic data and use it to train non-foundation and foundation models. Our approach has the potential to replace standard multi-dataset training of large time series models or serve as a complementary process. To generate data, we assume that the target distribution is mostly dominated by the fundamental target frequency and its harmonics. Thus, we propose to synthetically construct sinusoidal data, given the fundamental frequency (a scalar) of the target distribution. We derive the fundamental frequency using the sampling rate of the target dataset, which is typically given as a co-variate and exploited by several models (Cao et al., 2024; Liu et al., 2024a; Ekambaram et al., 2024; Liu et al., 2024c). See also App. B.1 for details on the relation between the sampling rate and the fundamental frequency, and ways to estimate it. To create our synthetic dataset, we first generate a pool of sines, followed by sampling from that pool and constructing various time series signals. We illustrate this approach in Fig. 3.

**A pool of sines.** Given the fundamental target frequency $\bar{\omega}$, we create a pool $P$ of size $m$ of sine waves whose frequencies are the harmonics of $\bar{\omega}$, i.e.,

$$P := \{s^1, s^2, \ldots, s^m\}, \quad \text{where,} \quad s_t^k := A_k \sin(2\pi t\, \omega_k + \phi_k), \quad k = 1, \ldots, m, \qquad (4)$$

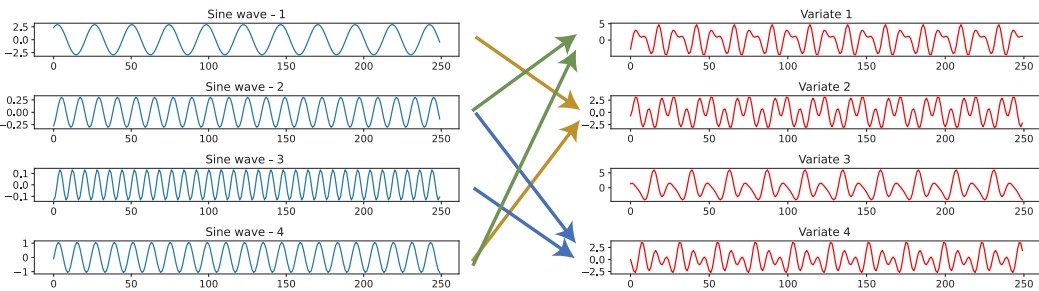

Figure 3: We construct a pool of sine waves that are harmonics to a given fundamental frequency (left). We create a multivariate time series by sampling and adding sines per variate (right).

where the amplitude is sampled from an exponential distribution $A_k \sim \text{Exp}(A')$, and the phase is drawn from a uniform distribution $\phi_k \sim \mathcal{U}([0, 2\pi])$. The frequencies $\omega_k$ are sampled from a uniform distribution over the harmonics. Namely, we have that $\omega_k \sim \mathcal{U}(\Omega)$, where $\Omega := \{\bar{\omega}, 2\bar{\omega}, \ldots, h\bar{\omega}\}$. The variables $m, h$ and $A'$ are hyper-parameters whose values are detailed in App. D.

**Dataset construction.** We generate a synthetic multivariate time series $x \in \mathbb{R}^{d \times n}$ of $d$ variates and length $n$, i.e., $x = (x_t^j)$, for $t = 1, \ldots, n$ and $j = 1, \ldots, d$, by sampling uniformly from $P$. In particular, we draw $l$ sine waves per variate $j$ and sum them together. Formally,

$$x^j = \sum_{i=1}^{l} s^i, \quad \text{where,} \quad s^i \sim \mathcal{U}(P), \quad j = 1, \ldots, d. \tag{5}$$

The hyper-parameter $l$ controls the number of sines sampled from $P$, and it directly influences the correlation factor in that if $l$ is larger, then more variates in the time series are correlated. To create a full dataset, we simply repeat the above process to generate multiple time series. For additional diversity, one can create several datasets with different parameters such as the number of harmonics.

## 5 EXPERIMENTS

In this section, we evaluate Freq-Synth in comparison to recent state-of-the-art (SOTA) forecasting approaches on three settings: zero-shot forecasting (Sec. 5.1), synthetic data comparison (Sec. 5.2), and few-shot forecasting (Sec. 5.3). We consider the popular long-term time series forecasting (LTSF) benchmark (Zhou et al., 2021; Wu et al., 2021), which uses data from multiple domains including energy, financial, weather, and traffic. We detail below training choices that are specific for each setting. The evaluation is performed on forecast horizons of $96, 192, 336$, and $720$. More details including implementations, hyper-parameters, and training process are described in App. D.

### 5.1 ZERO-SHOT FORECASTING

In this evaluation setting, we compare forecasting performances obtained by training on real-world data vs. training solely on synthetic data (generated with Freq-Synth). We consider the following SOTA baselines: TTM (Ekambaram et al., 2024), UniTime (Liu et al., 2024a), Moment (Goswami et al., 2024), Timer (Liu et al., 2024d), GPT4TS (Zhou et al., 2023), PatchTST (Nie et al., 2023). These models are either foundation models, designed for the zero-shot forecasting setting, or non-foundation models that marked significant milestones in developing such approaches. We also use the naive and seasonal-naive baselines for comparison. Inspired by TTM, we employ a similar setup and train these baselines in the real data case on a subset of datasets from Monash (Godahewa et al., 2021) and PEMS (Liu et al., 2022). These datasets include a large range of sampling rates, including 4 seconds, 10 minutes, 1 hour, and more. Importantly, all the sampling rates of the evaluation data are also contained in this large train set, except for 15 minutes, which is the sampling rate of the ETTm datasets. For the synthetic data case, we utilize Freq-Synth that comprises of three sine groups, corresponding to the harmonics $1, 2$, and $3$. Following the steps in Sec. 4.2, we sample only 5000 examples for training, and another 5000 data samples for validation. We emphasize that the

Table 1: A comparison of zero-shot forecasting when training with real data vs. synthetic data. Freq-Synth outperforms real data in the majority of cases. The full table is given in Tab.6.

| Model | TTM | | | | Timer | | | | UniTime | | | | Moment | | | |
|---|---|---|---|---|---|---|---|---|---|---|---|---|---|---|---|---|
| | Real | | Synth | | Real | | Synth | | Real | | Synth | | Real | | Synth | |
| | MSE | MAE | MSE | MAE | MSE | MAE | MSE | MAE | MSE | MAE | MSE | MAE | MSE | MAE | MSE | MAE |
| ETTh1 | 0.618 | 0.543 | **0.486** | **0.452** | 0.723 | 0.584 | **0.528** | **0.483** | 0.924 | 0.647 | **0.507** | **0.469** | 0.690 | 0.566 | **0.519** | **0.469** |
| ETTh2 | 0.415 | 0.422 | **0.412** | **0.420** | 0.604 | 0.492 | **0.482** | **0.454** | 0.552 | 0.488 | **0.419** | **0.424** | **0.410** | **0.423** | 0.418 | 0.424 |
| ETTm1 | 1.253 | 0.718 | **0.454** | **0.414** | 1.267 | 0.724 | **0.480** | **0.433** | 1.138 | 0.703 | **0.455** | **0.418** | 0.848 | 0.600 | **0.474** | **0.430** |
| ETTm2 | 0.398 | 0.412 | **0.328** | **0.346** | 0.413 | 0.422 | **0.352** | **0.363** | 0.400 | 0.416 | **0.334** | **0.348** | 0.328 | 0.366 | 0.337 | **0.350** |
| Exchange | **0.362** | **0.406** | 0.414 | 0.449 | **0.342** | **0.395** | 0.434 | 0.463 | **0.356** | **0.404** | 0.415 | 0.450 | 0.403 | 0.434 | 0.414 | 0.449 |
| Electricity | 0.427 | 0.494 | **0.280** | **0.354** | 0.511 | 0.528 | **0.302** | **0.368** | 0.575 | 0.563 | **0.285** | **0.360** | 0.768 | 0.716 | **0.294** | **0.370** |
| Traffic | 1.002 | 0.611 | **0.844** | **0.493** | 0.957 | 0.593 | **0.928** | **0.549** | 0.989 | 0.608 | **0.858** | **0.515** | 1.332 | 0.765 | **0.891** | **0.535** |
| Weather | **0.341** | **0.318** | 0.410 | 0.338 | **0.340** | **0.313** | 0.344 | 0.324 | **0.304** | **0.305** | 0.339 | 0.318 | **0.284** | **0.310** | 0.338 | 0.326 |
| Average | 0.602 | 0.490 | **0.454** | **0.408** | 0.645 | 0.506 | **0.481** | **0.430** | 0.655 | 0.517 | **0.452** | **0.413** | 0.633 | 0.522 | **0.461** | **0.419** |

| Model | GPT4TS | | | | PatchTST | | | | Naive | | S-Naive | |
|---|---|---|---|---|---|---|---|---|---|---|---|---|
| | Real | | Synth | | Real | | Synth | | | | | |
| | MSE | MAE | MSE | MAE | MSE | MAE | MSE | MAE | MSE | MAE | MSE | MAE |
| ETTh1 | 0.620 | 0.540 | **0.456** | **0.446** | 1.894 | 0.834 | **0.463** | **0.445** | 1.323 | 0.738 | 0.600 | 0.480 |
| ETTh2 | 0.528 | 0.475 | **0.401** | **0.414** | 0.623 | 0.517 | **0.405** | **0.414** | 0.540 | 0.481 | 0.483 | 0.437 |
| ETTm1 | 1.189 | 0.707 | **0.426** | **0.412** | 1.255 | 0.734 | **0.430** | **0.409** | 1.271 | 0.698 | 0.489 | 0.421 |
| ETTm2 | 0.394 | 0.414 | **0.310** | **0.335** | 0.453 | 0.442 | **0.316** | **0.338** | 0.385 | 0.394 | 0.358 | 0.358 |
| Exchange | **0.382** | **0.418** | 0.414 | 0.448 | **0.350** | **0.399** | 0.414 | 0.448 | **0.341** | **0.390** | 0.348 | 0.396 |
| Electricity | 0.440 | 0.497 | **0.310** | **0.400** | 0.692 | 0.594 | **0.280** | **0.362** | 1.612 | 0.958 | 0.330 | **0.342** |
| Traffic | 0.983 | 0.631 | **0.810** | **0.500** | 0.966 | 0.577 | **0.798** | **0.477** | 2.765 | 1.088 | 1.161 | 0.480 |
| Weather | 0.313 | 0.310 | **0.284** | **0.301** | **0.317** | **0.314** | 0.350 | 0.317 | 0.352 | 0.320 | 0.395 | 0.357 |
| Average | 0.606 | 0.499 | **0.426** | **0.407** | 0.819 | 0.551 | **0.432** | **0.401** | 1.074 | 0.633 | 0.520 | 0.409 |

real data is $\approx 1000$ times larger in volume than our synthetic data, resulting in significantly higher time and memory requirements.

We show in Tab. 1 the zero-shot forecasting results on several datasets (rows) as obtained by various methods (columns). We report the mean squared error (MSE) and mean absolute error (MAE) metrics. Each result represents the average MSE/MAE over the forecast horizons $96, 192, 336, 720$ and three random seeds. Red and black boldface represent the lowest score in the row and the lowest score per model, respectively. The bottom row lists the average errors across all datasets. Notably, the vast majority of bold values appear on the 'Synth' columns, whereas Synth-Freq struggles with the Exchange and Weather datasets. Overall, Freq-Synth outperforms real data in **6/8** benchmark datasets, presenting lower MSE and MAE scores on average. Moreover, training solely on synthetic data exhibits the best MSE and MAE scores (marked in red), reinforcing the superiority of Freq-Synth option in **6/8** cases. We would like to highlight the scores of ETTm1 and ETTm2, presenting a notable reduction of **60%** and **16%** across all models on average, respectively. Recalling that the 15 minutes sampling rate is not available in Monash and PEMS, we suggest that these results indicate poor frequency generalization. Likewise, we argue that the above models also suffer from frequency confusion in the remaining datasets, implied by the performance gaps above.

## 5.2 COMPARING SYNTHETIC APPROACHES

The following evaluation setting compares the forecasting results for training solely on synthetic information. We consider several recent approaches for generating synthetic time series, including TimesFM (Das et al., 2024), ForecastPFN (Dooley et al., 2024), and KernelSynth (Ansari et al., 2024). TimesFM and KernelSynth have been using synthetic data originally to diversify real data for pre-trained models, whereas ForecastPFN trains only on synthetic data. We generate for ForecastPFN, TimesFM, and KernelSynth 500 channels, each of length 1024 to ensure diversity. In this experiment, we compare the following setups: 1) Known target sampling rate, which can be exploited in ForecastPFN and seasonal naive (S-Naive) as well as in Freq-Synth; and 2) Unknown target sampling rate, assuming no prior knowledge on the target domain. In the latter setup, we introduce a variant of Freq-Synth, named **Freq-Synth Natural**, that includes datasets with common natural frequencies such as $1/30, 1/7, 1/24, 1/60$. We also create another variant, **Freq-Synth Mix**, which is a dataset with random frequencies from the pool $P$. Importantly, the configuration we consider is similar to the original setup of the compared methods.

We detail in Tab. 2 the results, with the left and right blocks corresponding to known and unknown sampling rates, respectively. The MSE and MAE measures are averaged on a forecasting horizon of 96 across all six models (see Sec. 5.1). As in Tab. 1, red and black boldface values represent lowest MSE/MAE for each dataset and block respectively. Notably, a large performance difference

Table 2: Comparison between different synthetic data methods with the known target sampling rate (left block) and without it (right block). The mean over the datasets is given in the last row.

| | Known Sampling Rate | | | | | | | | Unknown Sampling Rate | | | | | | | | | | | |
| | Freq-Synth | | TimesFM | | ForecastPFN | | S-Naive | | Freq-Synth Nat | | Freq-Synth Mix | | Ker-Synth | | TimesFM | | ForecastPFN | | Naive | |
| | MSE | MAE | MSE | MAE | MSE | MAE | MSE | MAE | MSE | MAE | MSE | MAE | MSE | MAE | MSE | MAE | MSE | MAE | MSE | MAE |
|---|---|---|---|---|---|---|---|---|---|---|---|---|---|---|---|---|---|---|---|---|
| ETTh1 | **0.433** | **0.427** | 0.530 | 0.492 | 0.780 | 0.580 | 0.513 | 0.434 | **0.542** | **0.492** | 0.708 | 0.561 | 0.693 | 0.552 | 0.889 | 0.634 | 0.705 | 0.571 | 1.297 | 0.714 |
| ETTh2 | **0.352** | **0.377** | 0.375 | 0.378 | 0.641 | 0.486 | 0.391 | 0.380 | 0.363 | 0.388 | **0.355** | **0.388** | 0.359 | 0.389 | 0.418 | 0.419 | 0.531 | 0.458 | 0.432 | 0.422 |
| ETTm1 | **0.389** | **0.385** | 0.505 | 0.462 | 1.250 | 0.721 | 0.423 | 0.387 | **0.553** | **0.486** | 0.700 | 0.550 | 0.647 | 0.521 | 2.248 | 0.787 | 1.985 | 0.902 | 1.214 | 0.665 |
| ETTm2 | 0.235 | 0.290 | **0.209** | **0.280** | 0.286 | 0.357 | 0.263 | 0.301 | 0.239 | 0.308 | 0.231 | 0.308 | **0.225** | **0.301** | 0.288 | 0.348 | 0.383 | 0.419 | 0.267 | 0.328 |
| Electricity | **0.277** | **0.355** | 0.401 | 0.451 | 0.590 | 0.502 | 0.321 | 0.326 | **0.387** | **0.446** | 0.856 | 0.765 | 0.833 | 0.748 | 0.998 | 0.805 | 0.599 | 0.539 | 1.588 | 0.945 |
| Traffic | **0.888** | **0.521** | 1.009 | 0.600 | 1.363 | 0.709 | 1.217 | 0.497 | **1.125** | **0.640** | 1.424 | 0.811 | 1.402 | 0.808 | 1.823 | 0.931 | 1.454 | 0.774 | 2.714 | 1.077 |
| Weather | 0.275 | 0.271 | **0.234** | **0.269** | 0.264 | 0.297 | 0.349 | 0.333 | 0.240 | 0.279 | **0.216** | **0.272** | 0.238 | 0.285 | 0.404 | 0.330 | 0.395 | 0.363 | 0.259 | 0.254 |
| Average | **0.407** | **0.375** | 0.466 | 0.419 | 0.739 | 0.522 | 0.497 | 0.380 | **0.493** | **0.434** | 0.641 | 0.522 | 0.628 | 0.515 | 1.010 | 0.608 | 0.865 | 0.575 | 1.110 | 0.629 |

Table 3: Few-shot performance comparison by fine-tuning the models from Sec. 5.1 on a fraction of the target domain. The last row represents the average MSE and MAE values across datasets.

| Model | TTM | | | | Timer | | | | PatchTST | | | |
| | Real | | Synth | | Real | | Synth | | Real | | Synth | |
| | MSE | MAE | MSE | MAE | MSE | MAE | MSE | MAE | MSE | MAE | MSE | MAE |
|---|---|---|---|---|---|---|---|---|---|---|---|---|
| ETTh1 | 0.509 | 0.475 | **0.416** | **0.419** | 0.565 | 0.496 | **0.436** | **0.438** | 0.608 | 0.534 | **0.419** | **0.426** |
| ETTh2 | 0.332 | **0.365** | **0.331** | **0.365** | **0.310** | **0.354** | 0.346 | 0.378 | 0.343 | 0.365 | **0.324** | **0.359** |
| ETTm1 | 0.608 | 0.496 | **0.416** | **0.420** | 0.878 | 0.571 | **0.435** | **0.426** | 0.785 | 0.537 | **0.480** | **0.459** |
| ETTm2 | **0.189** | **0.270** | 0.193 | **0.270** | 0.205 | **0.280** | 0.201 | 0.281 | 0.195 | 0.277 | **0.190** | **0.269** |
| Electricity | 0.200 | 0.285 | **0.196** | **0.281** | 0.186 | **0.265** | **0.184** | 0.266 | **0.209** | **0.309** | 0.211 | 0.310 |
| Traffic | 0.547 | 0.357 | **0.531** | **0.347** | 0.490 | 0.315 | **0.482** | **0.306** | 0.523 | **0.342** | **0.522** | 0.354 |
| Weather | **0.178** | **0.223** | 0.203 | 0.245 | **0.191** | **0.220** | 0.191 | 0.240 | **0.171** | **0.214** | 0.198 | 0.248 |
| Average | 0.366 | 0.353 | **0.327** | **0.335** | 0.404 | 0.357 | **0.325** | **0.334** | 0.405 | 0.368 | **0.335** | **0.346** |

is observed between the left and right blocks, in favor for the known sampling rate case. Thus, utilizing the target sampling rate or the fundamental frequency facilitates the alleviation of frequency confusion issues. When analyzing each block separately, we find Freq-Synth to be superior to the other baselines, presenting an MSE reduction of **12.7%** vs. TimesFM in the left block (0.407 vs. 0.466), and an **21.5%** reduction vs. KernelSynth in the right block (0.493 vs. 0.628). Remarkably, while Freq-Synth trains on a fraction (i.e., $1/14$) of the data ForecastPFN, TimesFM, and Synth-Freq use, it consistently obtains better error measures.

## 5.3 FEW-SHOT FORECASTING

We conclude this section with the few-shot evaluation setting, where a pre-trained model is allowed to fine-tune on the target domain with a limited number of examples. To this end, we utilize the TTM, Timer, and PatchTST models from Sec. 5.1 that forecast for a horizon of 96, and we fine-tune them on $10\%$ of the train and validation sets of the target domain. We present the results in Tab. 3, highlighting the effectivity of Freq-Synth even in this setting. Notably, many of the bold values arise in the 'Synth' columns. Particularly, in terms of average performance, Freq-Synth exhibits **10.6%**, **19.5%**, and **17.3%** overall MSE reduction for the models TTM, Timer, and PatchTST, respectively. These results suggest that with a lighter and more efficient synthetic setup, we can achieve competitive to better results with less training, and free of the associated disadvantages of real data such as data collection, cleaning, management and privacy issues.

# 6 ANALYSIS

## 6.1 PRE-TRAINED MODELS

Below, we further expand the discussion about frequency generalization and frequency confusion discussed in Sec. 4. Here, we consider pre-trained models, obtained from the original repositories of TimesFM (Das et al., 2024), Timer (Liu et al., 2024d), and TTM (Ekambaram et al., 2024). To test whether the given models can generalize well, we evaluated their performance on simple periodic signals, with one harmonic (sine wave) and two harmonics of different frequencies. The results are depicted in Fig. 4, where the left plots detail the test MSE in log scale as a function of the frequency, and the right plots show examples of the evaluated signals. We find that models achieve reasonable errors on the $1/24$ frequency and its 2-harmonic frequency $1/12$, where the red dashed lines are positioned. This could be explained by the amount of pre-training data associated with the $1/24$ frequency, which often relates to an hourly sampling rate. For example, hourly sampled data accounts for $> 62\%$ of the pre-training datasets of TimesFM (Das et al., 2024). On the other hand, when evaluated on less common frequencies, we observe a significant performance degradation with

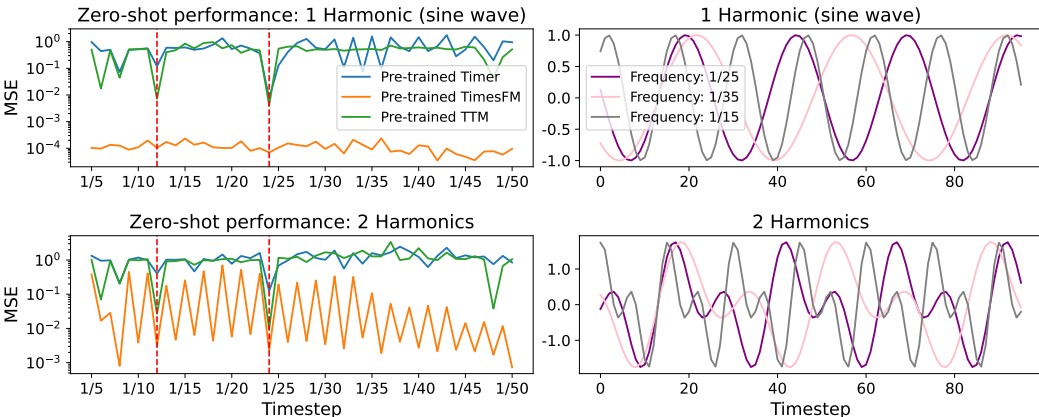

Figure 4: Zero-shot performance of pre-trained models on signals with one and two harmonics (top and bottom, respectively). The models perform well on the 1/24 and 1/12 frequencies, but for the remaining frequencies, the performance decreases significantly.

MSE values getting closer to 1 (left, top). This behavior becomes more apparent when the number of harmonics is greater than 1 (left, bottom). We further extend this evaluation in Fig. 9 with up to four harmonics. We also show the forecast predictions of individual signals for the over-fitted $1/24$ frequency in Fig. 7, and for the under-fitted $1/25$ frequency in Fig. 8. This analysis complements our frequency-based analysis above, suggesting that large time series forecasting models suffer from frequency confusion and attain poor frequency generalization.

## 6.2 Data Generation time

Generating synthetic data introduces overhead to the computation pipeline. In what follows, we compare the generation time of different synthetic approaches. We test this by generating one million time points comprised of 1000 variates each of length 1000 for each of the methods, Freq-Synth, TimesFM, ForecastPFN, and KernelSynth. The times each method needed are 0.1 seconds for Freq-Synth, 3 seconds for TimesFM, 14.6 seconds for ForecastPFN, and 138.2 minutes for KernelSynth, presenting a significant advantage to Freq-Synth (see inset). For each reported result, we calculated the average creation time of three datasets. In addition to the generation time, we also note that Freq-Synth is easy-to-code, requiring only a few lines of code as presented in the code snippet in App. 1.

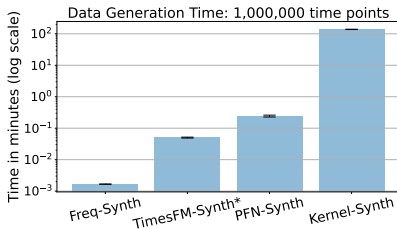

## 7 Conclusion

Deep foundation models are often considered for zero-shot and few-shot time series forecasting. However, the findings of this study emphasize the challenges these models face when dealing with complex frequency patterns and limited data availability. By employing Fourier analysis, we find that foundation and non-foundation models struggle to learn from multiple frequencies (frequency confusion), and exhibit limited generalization to frequencies that were unseen during training (poor frequency generalization). Toward addressing these challenging issues, we introduced *Freq-Synth*, a novel synthetic data generation framework that strategically enriches or replaces real data, based on the sampling rate of the target domain. Experimental results demonstrate that Freq-Synth improves the performance and robustness of both foundation and non-foundation models in zero-shot and few-shot learning settings. These contributions not only advance the understanding of frequency-based learning in time series forecasting but also offer a practical solution for enhancing model performance in low-data scenarios. Future research should further refine this approach by combining it with real data as well as further investigating its effects on foundation models.

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

Table 4: Freq-Synth hyper-parameters.

| Notation | Selected Value | Description |
|---|---|---|
| $A'$ | 5 | The expected amplitude for $A_k$ generation. |
| $\bar{\omega}$ | Depends on dataset, see B.1 | The estimated fundamental frequency |
| $m$ | 100 | Number of sine waves for the pool $P$ |
| $h$ | 1,2,3 | Determines the maximum number of harmonics |
| $l$ | 10 | The number of sine waves from $P$ used for the creation of $x^j$ |
| $n$ | 50,000 | A fixed length for the all sine waves and $x$ |
| $d$ | 5 | the number of variates for $x$ |

# A    APPENDIX

In this appendix, we provide additional information and details to supplement the main body of the paper. This includes supplementary method details in App. B, ablation studies in App. C, extended tables and results in App. E, and implementation details in App. D that were not included in the main text. The purpose of this appendix is to provide readers with a more comprehensive understanding of the research methodology and results, as well as to offer further insights into the experimental procedures and analysis.

# B    SUPPLEMENTARY METHOD DETAILS

## B.1    FUNDAMENTAL FREQUENCY ESTIMATION

The proposed Freq-Synth relies on prior information to overcome potential frequency confusion and improve frequency generalization. In this section we list different methods for obtaining this information required by our method.

**Sampling rate** information provides us with insight into the length between each collected time-step. This information can be further utilized to estimate the underlying fundamental frequency of the signal by bridging it to a common frequency. Common frequencies act as anchors, and attract a lot of valuable information due to their associations with strong natural or behavioral periods, namely, days, weeks, months, and years. For example, a behavioral signal could be linked to weekly or monthly patterns, as consumer behavior may vary on weekends or during specific months. This brings us to estimate the frequency based on the closest common frequency. For instance the sampling rates 5m, 10m, 15m, 30m, 1h are cycles of daily periods in which case the corresponding frequencies would be 1/288, 1/144, 1/96, 1/60, 1/48 and 1/24, respectively. A daily sampling rate is linked to a weekly 1/7 or monthly 1/30 (in our experiments we use a weekly rate). This continues for as long as a strong common frequency engulfs the sampling rate. This method is not free of challenges, for example the sampling rate may not be linked to the fundamental in cases where the fundamental frequency is unnatural, e.g. 1/100. Nevertheless, it may serve as a useful tool for frequency estimation.

**Periodogram** is a useful tool to estimate the spectral density of a signal (Schuster, 1898), given a sample series or a handful of samples one can utilize FFT to obtain the spectral density estimation. Given the spectral density estimation, the fundamental frequency is the lowest dominant frequency which is accompanied by harmonics (higher frequencies that are a positive integer multiple of the fundamental frequency), hence the periodogram has the potential to provide us with harmonics information as well as the fundamental frequency.

**Prior frequency information**, although not always intuitive, is another way of determining the fundamental frequency, this could be domain knowledge in a certain field as well as thorough analysis of the spectral density.

## B.2    SYNTH-FREQ HYPER-PARAMETERS AND IMPLEMENTATION DETAILS

In what follows, we complement the description in Sec. 4 and provide details and descriptions of the parameters and their selected values presented in Tab. 4.

To implement Synth-Freq, we follow the next steps: 1) Create three datasets, each corresponding to $h = 1, 2, 3$ with the given parameters in Tab. 4. To create each dataset, it is recommended to use the

code in App. 1 with the relevant parameters. 2) Each channel is standardized, according to the LTSF protocol (Nie et al., 2023; Wu et al., 2021; Zhou et al., 2021), however, this may not be required depending on the use case or the model, since many models include instance normalization (Nie et al., 2023) as part of their architectural pipeline. 3) From all three comprised datasets, we sample all together 5,000 samples each of the lookback and horizon of interest. In our implementation, we used a lookback of 96 and a horizon of 720, hence a sample length of 816 was used for training. The reason we employ sampling is due to the stationarity of each $x^j$ in $x$, where the same patterns are repeated along the entire signal, concluding that the entire signal's length is not necessary.

### B.3 FREQ-SYNTH IMPLEMENTATION

Here, we provide the python implementation for Freq-synth in Listing 1, covering the two steps described in Sec. 4.2.

```python
import numpy as np

# create the pool of sine waves step 1
def create_pool(m, n, A_avg, harmonics, w_fund):
    set_fund = [w_fund*i for i in range(1, harmonics+1) if w_fund*i<0.5]
     # fundamental set
    P = []  # pool of sine waves
    t = np.arange(n)  # timesteps
    for k in range(m):
        A = np.random.exponential(scale=A_avg-0.01) + 0.01  # to avoid 0
        w = np.random.choice(set_fund)
        phi = np.random.uniform(0, 2*np.pi)
        s_k = A * np.sin(2*np.pi*t*w + phi)
        P.append(s_k)
    return np.array(P)

# create the signal step 2
def create_synth(P, var, p_frac):
    m,n = P.shape  # number of sine waves
    l = int(m * p_frac)  # number of sine waves for sampling
    X = []  # Freq-Synth dataset
    for i in range(var):
        idx = np.random.choice(m, l)
        s_i = np.sum(P[idx], axis=0)
        X.append(s_i)
    return np.array(X)

A_avg = 1 # average amplitude
w_fund = 1/24 # fundamental frequency
harmonics = 3  # harmonics

var = 5 # number of variates
n = 250 # signal length
m = 100 # pool size
p_frac = 0.1 # determines the size of l, as a fraction of the pool size

P = create_pool(m, n, A_avg, harmonics, w_fund)
X = create_synth(P, var, p_frac)
```

Listing 1: Python code for the Freq-Synth method.

### B.4 FREQ-SYNTH LIMITATIONS

Unfortunately Freq-Synth is not always effective for all zero-shot scenarios. In cases where the there is a wider range of dominant frequencies, often leading to a signal with a higher degree of randomness (Demirel & Holz, 2024), capturing a single or even a handful of fundamental frequencies becomes challenging. In this particular case a single fundamental frequency based approach is not sufficient to represent the signal, and perhaps a mixed Freq-Synth approach is more suitable, as in Tab. 2. A particular case with the Exchange dataset is further discussed in App.E.1.

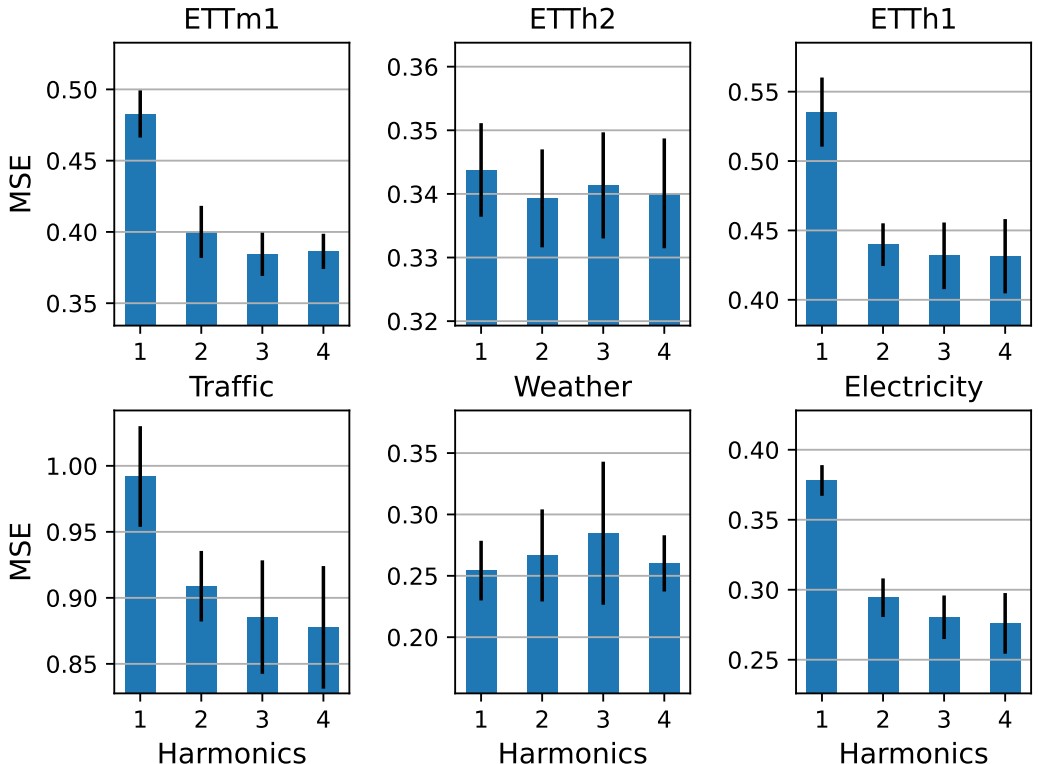

Figure 5: The influence of the number of harmonics on the ZS performance per dataset, where each result reports the average MSE for TTM, GPT4TS, PatchTST, UniTime, and Moment for a forecast horizon 96.

## C   ABLATION

### C.1   ABLATION: HARMONICS

We conduct experiments exploring the impact of different harmonics on the performance, showing the results in Fig. 5. In this experiment, we created four Freq-Synth configurations, following the steps in App. B, each with a fixed separate maximum number of harmonics corresponding to 1, 2, 3, and 4. It is shown that introducing $h > 1$ improves performance in all cases except for Weather, where the results are mixed. A significant improvement is given in ETTm1, ETTh1, Traffic and Electricity with an approximate reduction of 0.1 in the average MSE values. Introducing harmonics aids the model with focusing on higher periodic patterns that might be present in the data. For example, an hourly sampled signal which is associated with the daily period might also include semi-daily periods, e.g., night/day. Regarding $h > 2$, a small improvement is also visible in most datasets, as it enables the model to focus on smaller fine-grained periods. Their significance with respect to the MSE is however smaller due to the dominance of the larger periods, namely the fundamental frequency at $h = 1$.

### C.2   ABLATION: TRAINING DATA SIZE AND NUMBER OF VARIATES

We aim Freq-Synth to mimic the structure of real datasets such as Weather, ETT, Traffic and others. Therefore, in Freq-Synth the rate of correlation between synthetic variates is adjustable with the parameter $l$, as many datasets also include different rates of variate correlations. In Freq-Synth, we set the number of variates $d = 5$ and employ a single dataset size of 5,000, considering only the minimal shapes for the decision making. Nevertheless, in this ablation we test the effect of the dataset size and the number of variates on performance, the results are presented in Fig. 6. The results suggest that in most cases a number of variates greater than 1 is preferable with a lower MSE

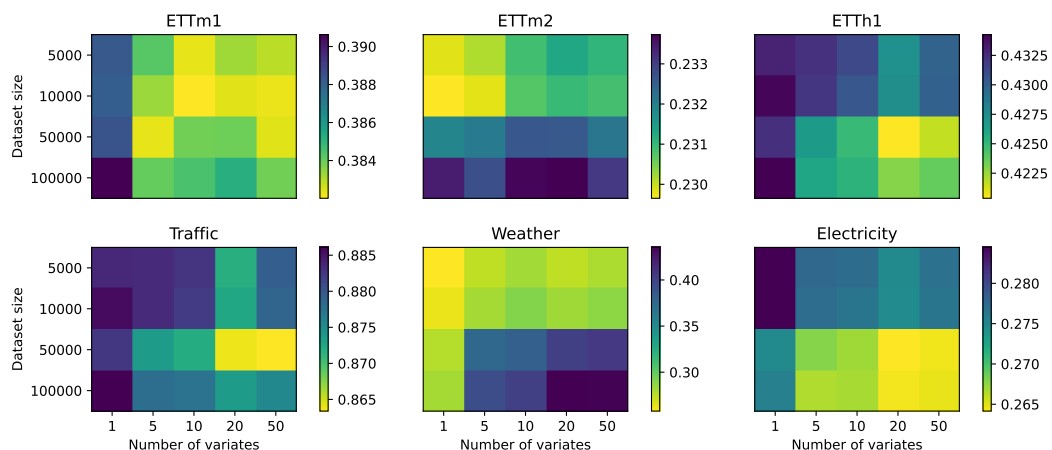

Figure 6: Ablation of dataset size and number of variates. The colorbar represents the average MSE of the models TTM, GPT4TS, PatchTST, UniTime, and Moment for a forecast horizon 96.

for each dataset size. For ETTm2 and Weather the results show otherwise, however, for dataset sizes 10,000 and 5,000, a comparable alternative is given for some $d > 1$. With respect to the dataset size, our experiments suggest an unclear pattern. For Weather, ETTm1, ETTm2, a smaller dataset size is preferable and the opposite for the remaining datasets. Therefore, we conclude that there is no clear trend regarding the effect of dataset size on performance.

## D   IMPLEMENTATION DETAILS

In this section, we provide additional details regarding the experimental settings, models, and datasets. Each experiment was carried out three times with three different random seeds to ensure robustness and reliability of the results. Our objective was to maintain fidelity to the original param-eters of each model, while establishing a unified framework for consistent comparison. Therefore, for each model we employ the original implementation with slight modifications to allow a fair com-parison in a unified framework. Throughout the experiments, a lookback of 96 and a train,test and validation fraction split of 0.6,0.2,0.2 respectively for ETT datasets and 0.7,0.2,0.1 for the remaining datasets was employed for training in accordance with the original protocol for the LTSF (Informer) datasets (Zhou et al., 2021; Wu et al., 2021; Liu et al., 2024b). The reported results represent the test fraction of the data. As for the pre-trained models in Figs. 4, 9, 7, and 8 lookback values of 512 for TTM and TimesFM, and 672 for Timer were utilized, to align with the specifications of the original trained models available online. All experiments were conducted with NVIDIA V100 32GB GPU, and each experiment was trained end to end on a single GPU.

### D.1   MODELS

In this work we selected the following models for evaluation:

- **PatchTST** (Nie et al., 2023).   An in-domain TSF transformer-based model, introduc-ing instance normalization, patching, a simple vanilla transformer and linear projection. PatchTST is a notable model, as many later released large-scale TSF models employ similar components including instance normalization, linear projection, patching, patch masking and reconstruction.
- **GPT4TS** (Zhou et al., 2023). A unified time-series model designed for a range of tasks in-cluding forecasting. GPT4TS uses a pre-trained frozen GPT-2 model, under the assumption that language domain data could be adapted to time-series data. GPT4TS is an important milestone towards foundation models as it showed success with employing a unified pre-trained language transformer for a range of downstream tasks with fine-tuning.
- **TTM** (Ekambaram et al., 2024) is a pre-trained model with a light-weight architecture which utilizes diverse resolution sampling with the implementation of patches of different

lengths and resolution prefix tuning, allowing the model to encode sampling rate specific information.

- **Timer** (Liu et al., 2024d) employs a GPT-style architecture, originally designed for a range of tasks such as imputation, anomaly detection, and forecasting. Although originally designed for auto-regressive next token prediction, we employ a non-auto regressive setup in this work, aligning our evaluation with other models.
- **Moment** (Goswami et al., 2024). A transformer-based foundation model for time-series, designed for various downstream tasks such as forecasting, classification, imputation and anomaly detection. Moment utilizes transformers, patching, and learnable mask embedding.
- **UniTime** (Liu et al., 2024a). A cross-domain large forecasting model empowered by a trainable GPT-2 backbone. UniTime also employs masking for generalization and to increase convergence speed. Language prompts are also utilized for identification information for training. However, we find in our implementation that this contribution hurts performance in ZS, therefore we do not provide "domain-instructions" as suggested in the original paper.

Although the given models offer a limited scope with respect to the available models, we selected these baselines for several reasons: 1) code availability for training: an easy access to trainable, original implementations through Hugging Face or github. 2) Performance and time efficiency: these works offer a thorough comparison to other comparable methods and showed better overall performance including faster inference or training time. For example, TTM's superior inference speed compared to other models (Ekambaram et al., 2024). 3) Prominence: for example PatchTST and GPT4TS are important milestones towards foundation models for TSF, due to their popularity and architectural contributions. These reasons eventually guided our decision making toward model selection for evaluation.

## D.2 DATASETS

In this work, we evaluate the proposed Freq-Synth on the common LTSF (Zhou et al., 2021) benchmark datasets. We train the baseline models in Tabs. 1 and 3 on a subset of datasets from the Monash repository (Godahewa et al., 2021). Specifically, we select the datasets that have a minimum length of 1,000 timesteps, in order to enable a training configuration of horizon length 720, which requires each example to be 846 timeseps long. The PEMS repository (Liu et al., 2022) is also included for training. This training setup is similar to the one employed in (Ekambaram et al., 2024). In Tab. 5, we provide details regarding the selected datasets for training and testing. To ensure that certain large datasets do no dominate training, we limit the maximum number of examples per dataset to 500,000 for training and validation. Selecting a subset of the entire training set is also a common practice in large unified training frameworks (Ekambaram et al., 2024; Liu et al., 2024d). In our case, it can prevent large datasets with many examples to engulf the effect of smaller and medium size datasets during training. The given train datasets cover a range of sectors such as nature, energy, traffic and financial and various sampling rates. Most sectors and sampling rates of the evaluation datasets are included in the train data, except for the sampling rate for ETTm1 and ETTm2.

## D.3 SYNTHETIC DATASETS

In what follows, we provide additional details regarding the synthetic datasets discussed in Sec. 5.2.

- **TimesFM** (Das et al., 2024): Synthetic generated data, where each channel selects up to three possible components that are eventually added together, or multiplied (trend only) among ARMA process, mixture of cosines and sines, and piece-wise linear trends. In this work, we provided results based on our implementation as the original implementation is not available.
- **ForecastPFN** (Dooley et al., 2024): They assume that there exists a shared distribution among real time-series datasets, which can be derived from natural periodic data, trend, global trends and noise. ForecastPFN synthetic data applies multiplication and addition to create signals that meet their prior distributions criteria. To handle extreme scales in their generated data, they introduce robust scaling and outlier removal, which is also employed for ForecastPFN in Tab. 2.

Table 5: Details on the considered datasets.

| Dataset | Repository | Channels | Min/max channel length | Sampling rate | Sector | Usage |
|---|---|---|---|---|---|---|
| ETTh1 | LTSF | 7 | 17,420 | hourly | Energy | Evaluation |
| ETTh2 | LTSF | 7 | 17,420 | hourly | Energy | Evaluation |
| ETTm1 | LTSF | 7 | 69,680 | 15 minutes | Energy | Evaluation |
| ETTm2 | LTSF | 7 | 69,680 | 15 minutes | Energy | Evaluation |
| Electricity | LTSF | 321 | 26,304 | hourly | Energy | Evaluation |
| Traffic | LTSF | 862 | 17,544 | hourly | Transport, Environmental | Evaluation |
| Weather | LTSF | 21 | 52,696 | 10 minutes | Nature | Evaluation |
| Exchange | LTSF | 8 | 7,588 | daily | Financial | Evaluation |
| London Smart Meters | Monash | 5,560 | 288/39,648 | 30 minutes | Energy | Training |
| Aus. Electricity Demand | Monash | 5 | 230,736/232,272 | 30 minutes | Energy, Environmental | Training |
| Wind Farms | Monash | 339 | 6,345/527,040 | minutely | Energy | Training |
| Bitcoin | Monash | 18 | 2,659/4,581 | daily | Financial | Training |
| KDD Cup 2018 | Monash | 270 | 9,504/10,920 | hourly | Nature, Environmental | Training |
| Weather (Monash) | Monash | 3,010 | 1,332/65,981 | daily | Nature | Training |
| Solar | Monash | 137 | 52,560 | 10 minutes | Nature | Training |
| Sunspot | Monash | 1 | 23,741 | daily | Nature | Training |
| Us Births | Monash | 1 | 7,305 | daily | Nature | Training |
| Saugeen River Flow | Monash | 1 | 23,741 | daily | Nature | Training |
| Solar Power | Monash | 1 | 7,397,222 | 4 seconds | Energy | Training |
| Wind Power | Monash | 1 | 7,397,147 | 4 seconds | Energy | Training |
| PEMS03 | PEMS | 358 | 25,887 | 5 minutes | Transport | Training |
| PEMS04 | PEMS | 307 | 16,992 | 5 minutes | Transport | Training |
| PEMS07 | PEMS | 883 | 28,224 | 5 minutes | Transport | Training |
| PEMS08 | PEMS | 170 | 17,856 | 5 minutes | Transport | Training |

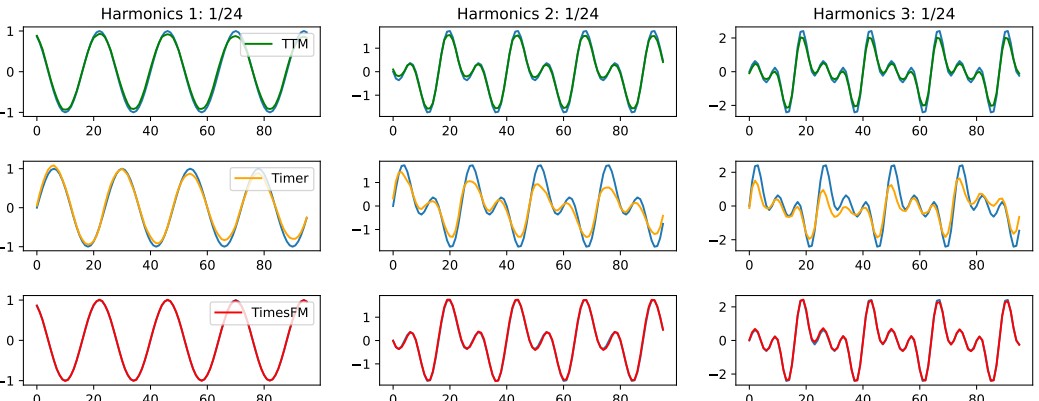

Figure 7: Forecast depiction of a 1/24 periodic series with different harmonics

- **KernelSynth** (Ansari et al., 2024): a Gaussian process (GP)-based synthetic time series generation method. Kernels are sampled from a kernel bank and then randomly combined using a binary operator ($\times$ or +). The resultant kernel is used in a GP prior to the generation of a synthetic time series.

In each of these methods, the generated channels are independent of the other channels, yet they attain cross-channel relations such as correlations and causality due to the underlying generation process. Freq-Synth on the other hand, supports multivariate channels with a controllable degree of linear similarly (Pearson correlation) through the parameter $l$. In Tab. 2, each synthetic dataset was standardized per channel except for ForecastPFN which was scaled with a robust scaler in accordance with the original implementation.

# E  EXTENDED EXPERIMENTS AND RESULTS

In this section, we provide additional depictions and tables that expand the experiments in the main body. The Fig. 10 depicts the Pearson correlation coefficient (PCC) between every pair of datasets included in Fig. 2. Values closer to 1 represent a higher Periodogram similarity.

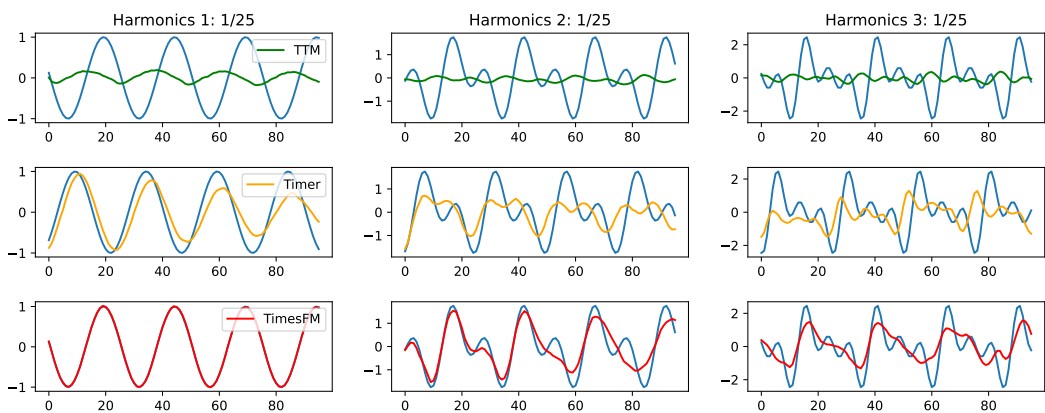

Figure 8: Forecast depiction of a 1/25 periodic series with different harmonics

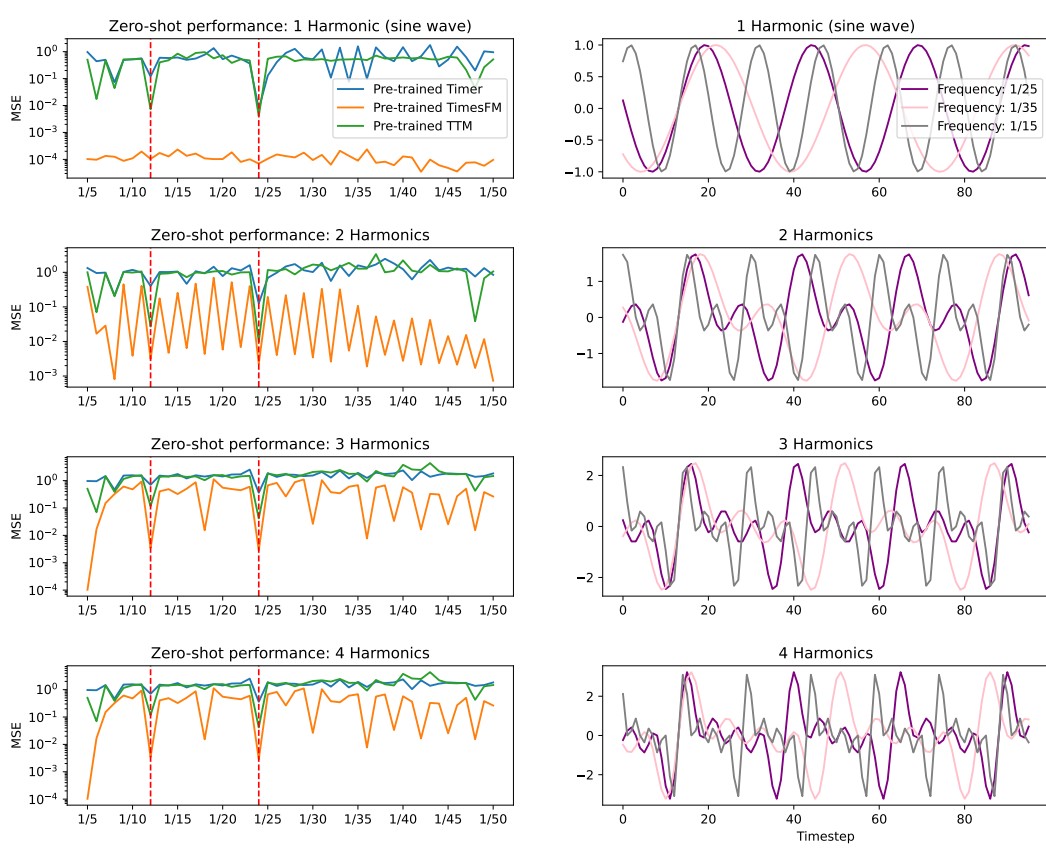

Figure 9: Zero-shot performance of pre-trained models on signals with 1 to 4 harmonics. The models perform well on the 1/24 and 1/12 frequencies, but for the remaining frequencies, the performance decreases significantly.

### E.1 PERIODOGRAM

The periodogram is often mention in this work as an effective tool for the analysis of a signal to extract the fundamental frequency. In Fig. 11, we provide examples of the periodograms of Exchange, Traffic and Electricity, where it is shown that the periodogram of Traffic and Electricity are very similar, with an identical fundamental frequency 1/24, with apparent harmonics, suggesting that they are potential candidates for successful transfer learning. Traffic and Electricity are both mentioned in the high periodogram correlation category in Fig. 2, and also shown in the periodogram correla-

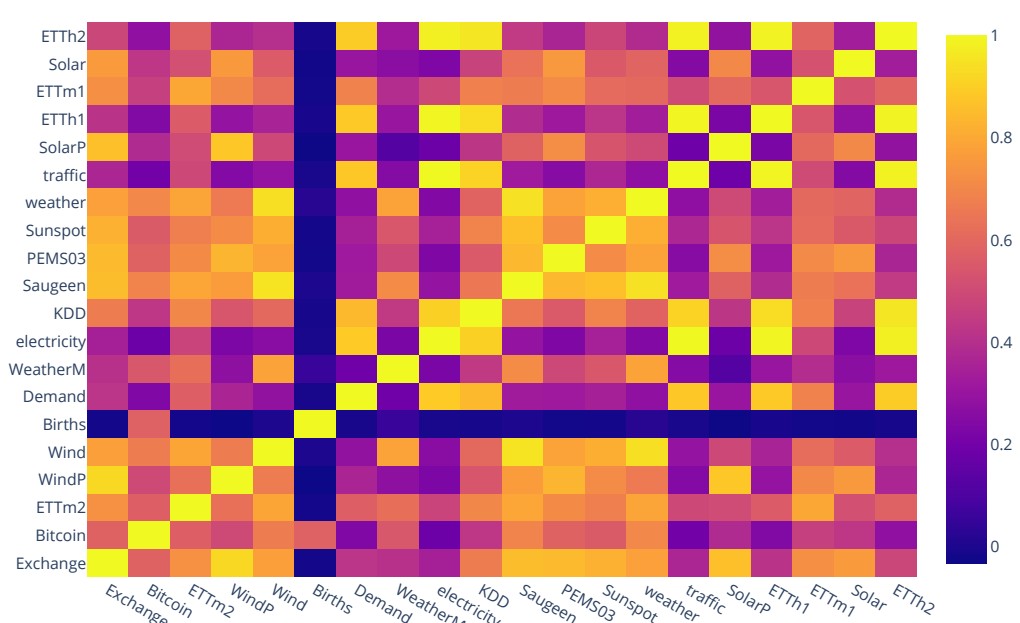

Figure 10: Pearson correlation score between the periodograms of each pair.

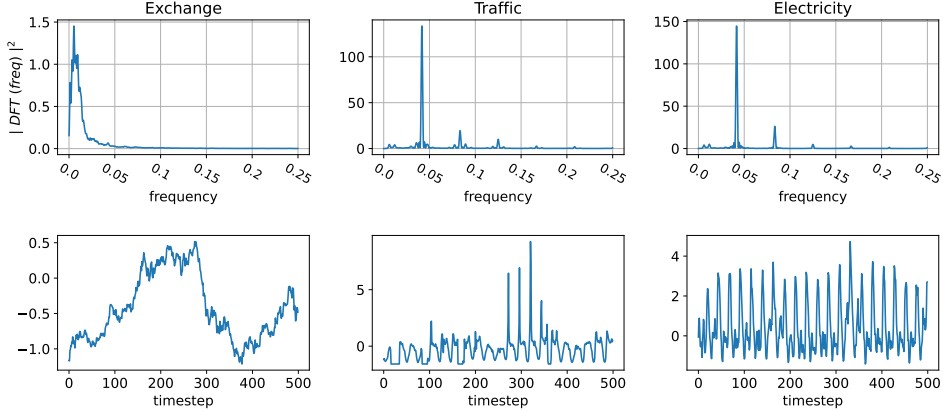

Figure 11: Top: Periodogram of the frequency range $(0, 0.25]$ for the datasets Exchange, Traffic, and Electricity visualized from left to right. Bottom: A random example from each dataset.

tion matrix Fig. 10 with PCC $> 0.9$. On the other hand, Exchange has a wider spread of significant frequencies, e.g., $(0, 0.025]$, hence, being characterized with more randomness without clear dominant fundamental frequencies. Consequently, Freq-Synth fails to perform well on Exchange, since the estimated fundamental frequency does not represent the true spectral span of the data.

Table 6: Zero-shot performance comparison of two setups: 1) Training only with Fq-Synth with target dataset sampling rate information. 2) Training with Real data from Monash and PEMS repositories. Each recorded result represents an average of three random seeds. Red and black bolds represents lowest score in the line and lowest score per model respectively.

| | Model | TTM Real MSE | TTM Real MAE | TTM Synth MSE | TTM Synth MAE | Timer Real MSE | Timer Real MAE | Timer Synth MSE | Timer Synth MAE | UniTime Real MSE | UniTime Real MAE | UniTime Synth MSE | UniTime Synth MAE | Moment Real MSE | Moment Real MAE | Moment Synth MSE | Moment Synth MAE | GPT4TS Real MSE | GPT4TS Real MAE | GPT4TS Synth MSE | GPT4TS Synth MAE | PatchTST Real MSE | PatchTST Real MAE | PatchTST Synth MSE | PatchTST Synth MAE | Naive MSE | Naive MAE | S-Naive MSE | S-Naive MAE |
|---|---|---|---|---|---|---|---|---|---|---|---|---|---|---|---|---|---|---|---|---|---|---|---|---|---|---|---|---|---|
| ETTh1 | 96 | 0.537 | 0.488 | 0.425 | 0.417 | 0.670 | 0.544 | 0.459 | 0.448 | 0.617 | 0.539 | 0.444 | 0.435 | 0.641 | 0.530 | 0.458 | 0.437 | 0.548 | 0.491 | 0.404 | 0.413 | 0.694 | 0.577 | 0.407 | 0.410 | 1.297 | 0.714 | 0.513 | 0.434 |
| | 192 | 0.642 | 0.553 | 0.485 | 0.450 | 0.733 | 0.585 | 0.520 | 0.481 | 0.760 | 0.614 | 0.506 | 0.468 | 0.688 | 0.559 | 0.519 | 0.469 | 0.655 | 0.557 | 0.456 | 0.443 | 1.154 | 0.729 | 0.463 | 0.442 | 1.325 | 0.733 | 0.581 | 0.469 |
| | 336 | 0.650 | 0.558 | 0.525 | 0.465 | 0.737 | 0.593 | 0.565 | 0.496 | 0.931 | 0.655 | 0.546 | 0.482 | 0.706 | 0.573 | 0.558 | 0.482 | 0.638 | 0.547 | 0.492 | 0.459 | 2.795 | 1.005 | 0.501 | 0.458 | 1.332 | 0.747 | 0.651 | 0.501 |
| | 720 | 0.642 | 0.572 | 0.510 | 0.474 | 0.751 | 0.613 | 0.566 | 0.508 | 1.389 | 0.781 | 0.532 | 0.491 | 0.724 | 0.602 | 0.542 | 0.488 | 0.639 | 0.566 | 0.473 | 0.471 | 2.933 | 1.026 | 0.482 | 0.469 | 1.337 | 0.756 | 0.656 | 0.514 |
| ETTh2 | 96 | 0.326 | 0.362 | 0.340 | 0.371 | 0.517 | 0.439 | 0.415 | 0.411 | 0.460 | 0.430 | 0.347 | 0.376 | 0.321 | 0.365 | 0.348 | 0.376 | 0.444 | 0.417 | 0.330 | 0.365 | 0.531 | 0.455 | 0.333 | 0.365 | 0.432 | 0.422 | 0.391 | 0.380 |
| | 192 | 0.427 | 0.419 | 0.420 | 0.417 | 0.650 | 0.502 | 0.495 | 0.454 | 0.587 | 0.497 | 0.428 | 0.422 | 0.416 | 0.419 | 0.427 | 0.422 | 0.555 | 0.482 | 0.409 | 0.410 | 0.664 | 0.528 | 0.413 | 0.411 | 0.534 | 0.473 | 0.482 | 0.429 |
| | 336 | 0.455 | 0.450 | 0.445 | 0.441 | 0.602 | 0.501 | 0.512 | 0.473 | 0.553 | 0.499 | 0.452 | 0.445 | 0.452 | 0.449 | 0.451 | 0.445 | 0.536 | 0.488 | 0.434 | 0.435 | 0.619 | 0.526 | 0.438 | 0.436 | 0.597 | 0.511 | 0.533 | 0.466 |
| | 720 | 0.453 | 0.457 | 0.442 | 0.449 | 0.645 | 0.524 | 0.505 | 0.479 | 0.608 | 0.528 | 0.448 | 0.452 | 0.453 | 0.460 | 0.447 | 0.452 | 0.579 | 0.514 | 0.432 | 0.444 | 0.678 | 0.559 | 0.435 | 0.444 | 0.595 | 0.519 | 0.526 | 0.474 |
| ETTm1 | 96 | 1.368 | 0.720 | 0.386 | 0.378 | 1.231 | 0.695 | 0.413 | 0.398 | 1.069 | 0.659 | 0.390 | 0.383 | 0.930 | 0.616 | 0.406 | 0.394 | 1.247 | 0.702 | 0.369 | 0.380 | 1.129 | 0.691 | 0.368 | 0.375 | 1.214 | 0.665 | 0.423 | 0.387 |
| | 192 | 1.207 | 0.709 | 0.429 | 0.400 | 1.283 | 0.736 | 0.456 | 0.419 | 1.170 | 0.712 | 0.431 | 0.404 | 0.914 | 0.620 | 0.449 | 0.416 | 1.244 | 0.724 | 0.406 | 0.400 | 1.253 | 0.741 | 0.408 | 0.396 | 1.261 | 0.690 | 0.463 | 0.406 |
| | 336 | 1.214 | 0.713 | 0.463 | 0.420 | 1.239 | 0.709 | 0.488 | 0.439 | 1.157 | 0.706 | 0.463 | 0.424 | 0.798 | 0.582 | 0.482 | 0.435 | 1.097 | 0.683 | 0.435 | 0.418 | 1.259 | 0.722 | 0.438 | 0.415 | 1.287 | 0.707 | 0.496 | 0.426 |
| | 720 | 1.223 | 0.732 | 0.538 | 0.457 | 1.314 | 0.756 | 0.565 | 0.476 | 1.157 | 0.734 | 0.535 | 0.461 | 0.749 | 0.583 | 0.558 | 0.473 | 1.169 | 0.718 | 0.496 | 0.451 | 1.379 | 0.780 | 0.505 | 0.450 | 1.323 | 0.730 | 0.574 | 0.465 |
| ETTm2 | 96 | 0.328 | 0.372 | 0.233 | 0.289 | 0.314 | 0.370 | 0.257 | 0.307 | 0.297 | 0.361 | 0.240 | 0.292 | 0.233 | 0.318 | 0.243 | 0.293 | 0.301 | 0.364 | 0.216 | 0.278 | 0.351 | 0.386 | 0.222 | 0.281 | 0.267 | 0.328 | 0.263 | 0.301 |
| | 192 | 0.343 | 0.387 | 0.290 | 0.325 | 0.394 | 0.416 | 0.314 | 0.342 | 0.380 | 0.410 | 0.297 | 0.327 | 0.298 | 0.352 | 0.300 | 0.328 | 0.373 | 0.408 | 0.273 | 0.313 | 0.442 | 0.439 | 0.279 | 0.316 | 0.340 | 0.371 | 0.321 | 0.337 |
| | 336 | 0.412 | 0.421 | 0.347 | 0.359 | 0.418 | 0.424 | 0.371 | 0.376 | 0.405 | 0.417 | 0.353 | 0.361 | 0.345 | 0.374 | 0.356 | 0.362 | 0.397 | 0.413 | 0.328 | 0.348 | 0.440 | 0.440 | 0.335 | 0.351 | 0.412 | 0.410 | 0.376 | 0.370 |
| | 720 | 0.509 | 0.467 | 0.442 | 0.412 | 0.527 | 0.480 | 0.466 | 0.428 | 0.518 | 0.478 | 0.448 | 0.414 | 0.434 | 0.419 | 0.450 | 0.415 | 0.504 | 0.470 | 0.424 | 0.402 | 0.571 | 0.504 | 0.430 | 0.405 | 0.522 | 0.466 | 0.471 | 0.422 |
| Exchange | 96 | 0.087 | 0.207 | 0.142 | 0.272 | 0.096 | 0.216 | 0.163 | 0.291 | 0.099 | 0.220 | 0.143 | 0.273 | 0.114 | 0.237 | 0.142 | 0.272 | 0.106 | 0.225 | 0.141 | 0.271 | 0.096 | 0.216 | 0.140 | 0.270 | 0.081 | 0.197 | 0.086 | 0.205 |
| | 192 | 0.198 | 0.313 | 0.240 | 0.355 | 0.183 | 0.304 | 0.260 | 0.371 | 0.198 | 0.315 | 0.241 | 0.356 | 0.233 | 0.341 | 0.240 | 0.355 | 0.226 | 0.332 | 0.239 | 0.354 | 0.186 | 0.306 | 0.239 | 0.354 | 0.167 | 0.289 | 0.173 | 0.295 |
| | 336 | 0.345 | 0.423 | 0.388 | 0.456 | 0.311 | 0.402 | 0.409 | 0.470 | 0.328 | 0.413 | 0.389 | 0.457 | 0.378 | 0.446 | 0.388 | 0.456 | 0.356 | 0.430 | 0.387 | 0.456 | 0.316 | 0.406 | 0.388 | 0.456 | 0.306 | 0.398 | 0.312 | 0.403 |
| | 720 | 0.819 | 0.681 | 0.888 | 0.714 | 0.776 | 0.657 | 0.906 | 0.721 | 0.799 | 0.669 | 0.888 | 0.714 | 0.886 | 0.713 | 0.888 | 0.714 | 0.842 | 0.687 | 0.887 | 0.713 | 0.802 | 0.669 | 0.887 | 0.713 | 0.810 | 0.675 | 0.819 | 0.680 |
| Electricity | 96 | 0.399 | 0.473 | 0.265 | 0.340 | 0.442 | 0.497 | 0.286 | 0.353 | 0.406 | 0.481 | 0.270 | 0.345 | 0.685 | 0.674 | 0.280 | 0.357 | 0.419 | 0.481 | 0.295 | 0.389 | 0.414 | 0.486 | 0.264 | 0.348 | 1.588 | 0.945 | 0.321 | 0.326 |
| | 192 | 0.418 | 0.490 | 0.264 | 0.343 | 0.476 | 0.521 | 0.286 | 0.357 | 0.472 | 0.521 | 0.269 | 0.348 | 0.732 | 0.699 | 0.278 | 0.359 | 0.437 | 0.496 | 0.295 | 0.390 | 0.586 | 0.560 | 0.264 | 0.351 | 1.596 | 0.951 | 0.304 | 0.324 |
| | 336 | 0.424 | 0.493 | 0.276 | 0.354 | 0.455 | 0.503 | 0.299 | 0.368 | 0.607 | 0.584 | 0.281 | 0.359 | 0.765 | 0.714 | 0.290 | 0.370 | 0.430 | 0.492 | 0.307 | 0.400 | 0.690 | 0.596 | 0.277 | 0.361 | 1.618 | 0.961 | 0.327 | 0.343 |
| | 720 | 0.467 | 0.522 | 0.314 | 0.380 | 0.670 | 0.592 | 0.336 | 0.393 | 0.814 | 0.666 | 0.320 | 0.386 | 0.892 | 0.775 | 0.329 | 0.396 | 0.473 | 0.518 | 0.343 | 0.422 | 1.079 | 0.733 | 0.313 | 0.386 | 1.647 | 0.975 | 0.367 | 0.373 |
| Traffic | 96 | 1.016 | 0.624 | 0.881 | 0.503 | 0.957 | 0.584 | 0.958 | 0.556 | 0.972 | 0.593 | 0.895 | 0.525 | 1.251 | 0.732 | 0.930 | 0.545 | 1.006 | 0.640 | 0.834 | 0.508 | 0.934 | 0.564 | 0.827 | 0.486 | 2.714 | 1.077 | 1.217 | 0.497 |
| | 192 | 0.989 | 0.604 | 0.819 | 0.486 | 0.934 | 0.589 | 0.906 | 0.543 | 0.950 | 0.595 | 0.834 | 0.508 | 1.295 | 0.752 | 0.865 | 0.528 | 0.993 | 0.649 | 0.786 | 0.493 | 0.927 | 0.566 | 0.774 | 0.470 | 2.747 | 1.085 | 1.092 | 0.458 |
| | 336 | 0.967 | 0.591 | 0.822 | 0.486 | 0.947 | 0.592 | 0.909 | 0.544 | 0.989 | 0.615 | 0.836 | 0.509 | 1.329 | 0.765 | 0.867 | 0.529 | 0.952 | 0.616 | 0.794 | 0.494 | 0.952 | 0.575 | 0.780 | 0.471 | 2.789 | 1.094 | 1.150 | 0.475 |
| | 720 | 1.034 | 0.626 | 0.855 | 0.497 | 0.991 | 0.606 | 0.941 | 0.553 | 1.047 | 0.630 | 0.868 | 0.519 | 1.451 | 0.812 | 0.900 | 0.539 | 0.982 | 0.617 | 0.825 | 0.503 | 1.050 | 0.605 | 0.812 | 0.481 | 2.810 | 1.097 | 1.185 | 0.489 |
| Weather | 96 | 0.244 | 0.256 | 0.348 | 0.291 | 0.264 | 0.251 | 0.272 | 0.264 | 0.217 | 0.239 | 0.270 | 0.269 | 0.197 | 0.247 | 0.309 | 0.308 | 0.223 | 0.248 | 0.209 | 0.249 | 0.228 | 0.248 | 0.282 | 0.264 | 0.259 | 0.254 | 0.349 | 0.333 |
| | 192 | 0.307 | 0.299 | 0.389 | 0.350 | 0.318 | 0.295 | 0.312 | 0.303 | 0.275 | 0.286 | 0.311 | 0.300 | 0.255 | 0.292 | 0.309 | 0.308 | 0.293 | 0.297 | 0.251 | 0.281 | 0.288 | 0.294 | 0.303 | 0.299 | 0.309 | 0.292 | 0.354 | 0.331 |
| | 336 | 0.357 | 0.332 | 0.425 | 0.350 | 0.345 | 0.326 | 0.361 | 0.337 | 0.318 | 0.320 | 0.357 | 0.331 | 0.304 | 0.322 | 0.356 | 0.338 | 0.330 | 0.325 | 0.303 | 0.315 | 0.337 | 0.332 | 0.367 | 0.330 | 0.376 | 0.338 | 0.402 | 0.360 |
| | 720 | 0.456 | 0.387 | 0.476 | 0.389 | 0.433 | 0.380 | 0.431 | 0.382 | 0.404 | 0.374 | 0.419 | 0.373 | 0.382 | 0.377 | 0.420 | 0.380 | 0.405 | 0.374 | 0.374 | 0.360 | 0.415 | 0.384 | 0.426 | 0.372 | 0.465 | 0.394 | 0.477 | 0.404 |

Table 7: Full table of Tab. 2.

| | | Known Sampling Rate | | | | | | | | Unknown Sampling Rate | | | | | | | | | | | |
| | | Fq-Synth MSE | Fq-Synth MAE | FM MSE | FM MAE | PFN MSE | PFN MAE | S-Naive MSE | S-Naive MAE | Fq-Synth Nat MSE | Fq-Synth Nat MAE | Fq-Synth Mix MSE | Fq-Synth Mix MAE | Ker-Synth MSE | Ker-Synth MAE | FM MSE | FM MAE | PFN MSE | PFN MAE | Naive MSE | Naive MAE |
|---|---|---|---|---|---|---|---|---|---|---|---|---|---|---|---|---|---|---|---|---|---|
| PatchTST | ETTh1 | 0.407 | 0.410 | 0.496 | 0.469 | 0.816 | 0.591 | 0.513 | 0.434 | 0.640 | 0.536 | 0.709 | 0.562 | 0.700 | 0.553 | 0.587 | 0.541 | 0.589 | 0.541 | 1.297 | 0.714 |
| | ETTh2 | 0.333 | 0.365 | 0.358 | 0.372 | 0.683 | 0.507 | 0.391 | 0.380 | 0.389 | 0.404 | 0.355 | 0.388 | 0.400 | 0.405 | 0.359 | 0.392 | 0.478 | 0.435 | 0.432 | 0.422 |
| | ETTm1 | 0.368 | 0.375 | 0.475 | 0.445 | 1.204 | 0.713 | 0.423 | 0.387 | 0.555 | 0.493 | 0.704 | 0.552 | 0.816 | 0.579 | 1.133 | 0.638 | 1.740 | 0.852 | 1.214 | 0.665 |
| | ETTm2 | 0.222 | 0.281 | 0.200 | 0.273 | 0.276 | 0.353 | 0.263 | 0.301 | 0.252 | 0.316 | 0.231 | 0.308 | 0.251 | 0.316 | 0.235 | 0.312 | 0.353 | 0.401 | 0.267 | 0.328 |
| | Electricity | 0.264 | 0.348 | 0.375 | 0.446 | 0.561 | 0.468 | 0.321 | 0.326 | 0.505 | 0.510 | 0.857 | 0.766 | 0.857 | 0.746 | 0.911 | 0.773 | 0.490 | 0.479 | 1.588 | 0.945 |
| | Traffic | 0.827 | 0.486 | 0.952 | 0.569 | 1.231 | 0.658 | 1.217 | 0.497 | 1.457 | 0.766 | 1.426 | 0.811 | 1.416 | 0.818 | 1.548 | 0.872 | 1.288 | 0.715 | 2.714 | 1.077 |
| | Weather | 0.282 | 0.267 | 0.207 | 0.250 | 0.265 | 0.300 | 0.349 | 0.333 | 0.294 | 0.315 | 0.216 | 0.271 | 0.294 | 0.304 | 0.347 | 0.340 | 0.529 | 0.254 | 0.259 | 0.254 |
| | Average | 0.386 | 0.362 | 0.438 | 0.403 | 0.719 | 0.513 | 0.497 | 0.380 | 0.577 | 0.471 | 0.643 | 0.523 | 0.676 | 0.533 | 0.750 | 0.554 | 0.761 | 0.538 | 1.110 | 0.629 |
| GPT4TS | ETTh1 | 0.404 | 0.413 | 0.503 | 0.476 | 0.996 | 0.640 | 0.513 | 0.434 | 0.486 | 0.462 | 0.711 | 0.562 | 0.674 | 0.546 | 0.606 | 0.523 | 0.725 | 0.578 | 1.297 | 0.714 |
| | ETTh2 | 0.330 | 0.365 | 0.353 | 0.370 | 0.802 | 0.542 | 0.391 | 0.380 | 0.341 | 0.375 | 0.355 | 0.390 | 0.342 | 0.381 | 0.480 | 0.438 | 0.599 | 0.482 | 0.432 | 0.422 |
| | ETTm1 | 0.369 | 0.380 | 0.466 | 0.444 | 1.337 | 0.740 | 0.423 | 0.387 | 0.549 | 0.483 | 0.706 | 0.552 | 0.589 | 0.499 | 3.225 | 0.885 | 2.547 | 1.006 | 1.214 | 0.665 |
| | ETTm2 | 0.216 | 0.278 | 0.203 | 0.275 | 0.292 | 0.361 | 0.263 | 0.301 | 0.228 | 0.301 | 0.231 | 0.309 | 0.215 | 0.295 | 0.375 | 0.388 | 0.437 | 0.451 | 0.267 | 0.328 |
| | Electricity | 0.295 | 0.389 | 0.396 | 0.448 | 0.654 | 0.487 | 0.321 | 0.326 | 0.387 | 0.469 | 0.861 | 0.768 | 0.803 | 0.738 | 0.940 | 0.782 | 0.534 | 0.473 | 1.588 | 0.945 |
| | Traffic | 0.834 | 0.508 | 0.985 | 0.584 | 1.598 | 0.744 | 1.217 | 0.497 | 0.997 | 0.603 | 1.431 | 0.814 | 1.374 | 0.799 | 2.008 | 0.923 | 1.427 | 0.752 | 2.714 | 1.077 |
| | Weather | 0.209 | 0.249 | 0.229 | 0.266 | 0.265 | 0.295 | 0.349 | 0.333 | 0.227 | 0.275 | 0.217 | 0.273 | 0.219 | 0.274 | 0.766 | 0.412 | 0.466 | 0.388 | 0.259 | 0.254 |
| | Average | 0.380 | 0.369 | 0.448 | 0.409 | 0.849 | 0.544 | 0.497 | 0.380 | 0.459 | 0.424 | 0.645 | 0.524 | 0.602 | 0.505 | 1.234 | 0.633 | 0.961 | 0.590 | 1.110 | 0.629 |
| Moment | ETTh1 | 0.458 | 0.437 | 0.612 | 0.537 | 0.544 | 0.498 | 0.513 | 0.434 | 0.612 | 0.483 | 0.706 | 0.559 | 0.706 | 0.558 | 1.334 | 0.817 | 0.990 | 0.568 | 1.297 | 0.714 |
| | ETTh2 | 0.348 | 0.376 | 0.354 | 0.378 | 0.390 | 0.394 | 0.391 | 0.380 | 0.359 | 0.385 | 0.354 | 0.388 | 0.354 | 0.388 | 0.501 | 0.480 | 0.480 | 0.443 | 0.432 | 0.422 |
| | ETTm1 | 0.406 | 0.394 | 0.519 | 0.478 | 1.268 | 0.727 | 0.423 | 0.387 | 0.560 | 0.486 | 0.697 | 0.549 | 0.696 | 0.546 | 1.637 | 0.837 | 1.549 | 0.814 | 1.214 | 0.665 |
| | ETTm2 | 0.243 | 0.293 | 0.237 | 0.298 | 0.304 | 0.366 | 0.263 | 0.301 | 0.237 | 0.306 | 0.230 | 0.308 | 0.230 | 0.308 | 0.292 | 0.376 | 0.343 | 0.396 | 0.267 | 0.328 |
| | Electricity | 0.280 | 0.357 | 0.450 | 0.481 | 0.454 | 0.481 | 0.321 | 0.326 | 0.359 | 0.428 | 0.849 | 0.763 | 0.852 | 0.762 | 1.315 | 0.916 | 0.714 | 0.650 | 1.588 | 0.945 |
| | Traffic | 0.930 | 0.545 | 1.158 | 0.674 | 1.117 | 0.647 | 1.217 | 0.497 | 1.023 | 0.597 | 1.415 | 0.807 | 1.423 | 0.813 | 2.613 | 1.199 | 1.384 | 0.767 | 2.714 | 1.077 |
| | Weather | 0.268 | 0.278 | 0.274 | 0.292 | 0.265 | 0.297 | 0.349 | 0.333 | 0.217 | 0.273 | 0.217 | 0.273 | 0.217 | 0.273 | 0.353 | 0.337 | 0.354 | 0.350 | 0.259 | 0.254 |
| | Average | 0.419 | 0.383 | 0.515 | 0.448 | 0.620 | 0.487 | 0.497 | 0.380 | 0.471 | 0.424 | 0.638 | 0.521 | 0.640 | 0.521 | 1.149 | 0.709 | 0.789 | 0.570 | 1.110 | 0.629 |
| UniTime | ETTh1 | 0.444 | 0.435 | 0.540 | 0.500 | 0.861 | 0.599 | 0.513 | 0.434 | 0.544 | 0.492 | 0.706 | 0.560 | 0.696 | 0.552 | 0.755 | 0.583 | 0.589 | 0.519 | 1.297 | 0.714 |
| | ETTh2 | 0.347 | 0.376 | 0.487 | 0.413 | 0.932 | 0.568 | 0.391 | 0.380 | 0.363 | 0.388 | 0.354 | 0.388 | 0.356 | 0.389 | 0.410 | 0.407 | 0.531 | 0.446 | 0.432 | 0.422 |
| | ETTm1 | 0.390 | 0.383 | 0.515 | 0.463 | 1.232 | 0.723 | 0.423 | 0.387 | 0.554 | 0.481 | 0.697 | 0.549 | 0.612 | 0.509 | 2.786 | 0.828 | 1.916 | 0.888 | 1.214 | 0.665 |
| | ETTm2 | 0.240 | 0.292 | 0.207 | 0.277 | 0.284 | 0.361 | 0.263 | 0.301 | 0.239 | 0.306 | 0.230 | 0.308 | 0.222 | 0.298 | 0.271 | 0.329 | 0.370 | 0.410 | 0.267 | 0.328 |
| | Electricity | 0.270 | 0.345 | 0.418 | 0.451 | 0.683 | 0.507 | 0.321 | 0.326 | 0.358 | 0.417 | 0.836 | 0.763 | 0.838 | 0.749 | 0.926 | 0.774 | 0.490 | 0.470 | 1.588 | 0.945 |
| | Traffic | 0.895 | 0.525 | 0.989 | 0.594 | 1.420 | 0.727 | 1.217 | 0.497 | 1.066 | 0.611 | 1.417 | 0.808 | 1.416 | 0.816 | 1.506 | 0.841 | 1.228 | 0.680 | 2.714 | 1.077 |
| | Weather | 0.270 | 0.269 | 0.230 | 0.266 | 0.270 | 0.306 | 0.349 | 0.333 | 0.239 | 0.279 | 0.216 | 0.272 | 0.230 | 0.281 | 0.329 | 0.317 | 0.392 | 0.364 | 0.259 | 0.254 |
| | Average | 0.408 | 0.375 | 0.484 | 0.423 | 0.812 | 0.542 | 0.497 | 0.380 | 0.480 | 0.425 | 0.638 | 0.521 | 0.624 | 0.513 | 0.998 | 0.583 | 0.788 | 0.540 | 1.110 | 0.629 |
| TTM | ETTh1 | 0.425 | 0.417 | 0.518 | 0.488 | 0.796 | 0.594 | 0.513 | 0.434 | 0.511 | 0.479 | 0.716 | 0.564 | 0.687 | 0.552 | 0.867 | 0.635 | 0.697 | 0.577 | 1.297 | 0.714 |
| | ETTh2 | 0.340 | 0.371 | 0.359 | 0.375 | 0.672 | 0.509 | 0.391 | 0.380 | 0.362 | 0.388 | 0.366 | 0.401 | 0.389 | 0.413 | 0.574 | 0.473 | 0.473 | 0.432 | 0.432 | 0.422 |
| | ETTm1 | 0.386 | 0.378 | 0.500 | 0.459 | 1.323 | 0.746 | 0.423 | 0.387 | 0.554 | 0.488 | 0.705 | 0.551 | 0.573 | 0.491 | 3.348 | 0.878 | 2.399 | 0.987 | 1.214 | 0.665 |
| | ETTm2 | 0.233 | 0.289 | 0.202 | 0.277 | 0.287 | 0.362 | 0.263 | 0.301 | 0.241 | 0.311 | 0.231 | 0.308 | 0.216 | 0.293 | 0.308 | 0.359 | 0.431 | 0.446 | 0.267 | 0.328 |
| | Electricity | 0.265 | 0.340 | 0.386 | 0.440 | 0.569 | 0.476 | 0.321 | 0.326 | 0.338 | 0.409 | 0.871 | 0.771 | 0.815 | 0.742 | 0.980 | 0.807 | 0.604 | 0.543 | 1.588 | 0.945 |
| | Traffic | 0.881 | 0.503 | 1.003 | 0.597 | 1.371 | 0.699 | 1.217 | 0.497 | 1.123 | 0.639 | 1.444 | 0.819 | 1.385 | 0.802 | 1.740 | 0.910 | 1.491 | 0.802 | 2.714 | 1.077 |
| | Weather | 0.348 | 0.291 | 0.236 | 0.272 | 0.274 | 0.304 | 0.349 | 0.333 | 0.243 | 0.282 | 0.217 | 0.272 | 0.246 | 0.289 | 0.467 | 0.345 | 0.452 | 0.389 | 0.259 | 0.254 |
| | Average | 0.411 | 0.370 | 0.458 | 0.415 | 0.756 | 0.527 | 0.497 | 0.380 | 0.482 | 0.428 | 0.649 | 0.525 | 0.611 | 0.508 | 1.182 | 0.621 | 0.950 | 0.602 | 1.110 | 0.629 |
| Timer | ETTh1 | 0.459 | 0.448 | 0.508 | 0.482 | 0.670 | 0.557 | 0.513 | 0.434 | 0.551 | 0.499 | 0.702 | 0.559 | 0.693 | 0.555 | 0.766 | 0.577 | 0.889 | 0.646 | 1.297 | 0.714 |
| | ETTh2 | 0.415 | 0.411 | 0.338 | 0.362 | 0.366 | 0.395 | 0.391 | 0.380 | 0.365 | 0.391 | 0.354 | 0.388 | 0.351 | 0.385 | 0.357 | 0.387 | 0.527 | 0.468 | 0.432 | 0.422 |
| | ETTm1 | 0.413 | 0.398 | 0.554 | 0.481 | 1.137 | 0.676 | 0.423 | 0.387 | 0.547 | 0.483 | 0.694 | 0.548 | 0.597 | 0.504 | 1.359 | 0.657 | 1.760 | 0.867 | 1.214 | 0.665 |
| | ETTm2 | 0.257 | 0.307 | 0.204 | 0.279 | 0.340 | 0.340 | 0.263 | 0.301 | 0.237 | 0.307 | 0.230 | 0.307 | 0.217 | 0.294 | 0.250 | 0.323 | 0.364 | 0.411 | 0.267 | 0.328 |
| | Electricity | 0.286 | 0.353 | 0.380 | 0.439 | 0.617 | 0.592 | 0.321 | 0.326 | 0.373 | 0.441 | 0.846 | 0.762 | 0.831 | 0.753 | 0.917 | 0.779 | 0.762 | 0.619 | 1.588 | 0.945 |
| | Traffic | 0.958 | 0.556 | 0.968 | 0.582 | 1.442 | 0.778 | 1.217 | 0.497 | 1.082 | 0.627 | 1.411 | 0.805 | 1.395 | 0.801 | 1.525 | 0.841 | 1.908 | 0.927 | 2.714 | 1.077 |
| | Weather | 0.272 | 0.272 | 0.229 | 0.266 | 0.242 | 0.282 | 0.349 | 0.333 | 0.244 | 0.284 | 0.216 | 0.271 | 0.225 | 0.278 | 0.218 | 0.266 | 0.366 | 0.347 | 0.259 | 0.254 |
| | Average | 0.437 | 0.392 | 0.454 | 0.413 | 0.678 | 0.517 | 0.497 | 0.380 | 0.486 | 0.433 | 0.636 | 0.520 | 0.616 | 0.510 | 0.770 | 0.547 | 0.939 | 0.612 | 1.110 | 0.629 |

