# OpenReview forum: "Beyond Data Scarcity: A Frequency-Driven Framework for Zero-Shot Forecasting"
_ICLR.cc/2025/Conference — ICLR 2025 Conference Withdrawn Submission_

### Official Review · Reviewer_21jG · 2024-10-31

**Soundness:** 2
**Presentation:** 3
**Contribution:** 2
**Rating:** 3
**Confidence:** 4

**Summary:**

In scenarios that require zero-shot or few-shot prediction capabilities due to data sparsity, mainstream approaches often utilize time-series foundational models. The authors point out that current time-series foundational models not only demand a substantial amount of pretraining data and incur high computational costs, but they are also limited by the issue of frequency confusion, which restricts their generalization performance. In light of this, the authors propose Freq-synth, an innovative pretraining data synthesis technique that allows the model to achieve good generalization performance in downstream tasks with minimal synthetic data for pretraining, thereby reducing the high costs associated with pretraining.

**Strengths:**

- The authors identify a potential reason for the limited generalization performance of current time-series models through experimental methods: frequency confusion. This occurs when the training set includes irrelevant frequency information beyond the target frequency, leading to a decline in generalization performance.
- Based on the aforementioned observations, the authors propose a simple and effective data synthesis technique called Freq-synth. This method synthesizes a task specific pretraining dataset based on the fundamental frequency, thus effectively addressing the frequency confusion issue previously mentioned. By pretraining on a minimal amount of synthetic data, the model can achieve zero-shot and few-shot forecasting.

**Weaknesses:**

- The paper introduces the concept of frequency generalization, which refers to the model's ability to perform well on unseen frequencies. However, Freq-synth essentially estimates the frequencies of the target dataset and synthesizes corresponding data based on those frequencies, which does not resolve the issue of frequency generalization.
- Freq-Synth generates task-specific synthetic data and uses it to train non-foundation and foundation models. Therefore, Freq-Synth essentailly still needs to train foundation models according to task specific information. Although it does not use the task specific data itself to train the foundation models, it more or less needs the information from the task data. I wouldn't call this full-shot training, but I wouldn't call it purely zero-shot of foundation models, either. If the foundation models need to be trained on task specific information, I would rather call them task specific models. This contradicts to the design philosophy of foundation models that need to see data from multiple domains that are also unseen during zero-shot.
- A serious issue with the paper is the significant discrepancy between the paper's experimental results and those reported in original literatures. In Section 5.1, the authors pretrain all models on both Real and Synth datasets and then compare zero-shot prediction performance in downstream tasks.  For example, the Real dataset contains most of the pretraining data used for TTM. However, in this experiment, TTM's performance after pretraining on the Real dataset is significantly worse than what is presented in its original paper. This seriously undermines the credibility of the study. This also applies to other baselines, whose results in the original papers show have shown better performance.
- In Section 5.2, the paper introduces Freq-synth Mix and mentions in the limitations that this method is better suited for data with complex signals. This raises a question about whether this aligns with the concept of frequency confusion. If Freq-synth Mix is designed for complex signals, it may inadvertently introduce more irrelevant frequency information, potentially exacerbating frequency confusion rather than mitigating it. This apparent contradiction warrants further discussion.
- The paper misses several SOTA baselines that could perform zero-shot, including foundation models (e.g., TimesFM, UniTS, MOIRAI) and LLM-based models (e.g., S^2IP-LLM, UniTime, Time-LLM). The specific model PatchTST is a quite old baseline, and more recent baselines, such as FITS, iTransformer, TimesNet, TimeMixer, should be included.

**Questions:**

See weaknesses.

---

### Official Review · Reviewer_QUJP · 2024-11-01

**Soundness:** 2
**Presentation:** 2
**Contribution:** 3
**Rating:** 5
**Confidence:** 4

**Summary:**

Existing foundation models may not have effectively learned from large amounts of training data, resulting in poor forecasting performance in zero-shot and few-shot scenarios. This paper highlights that frequency is a key factor influencing effective learning from data. It suggests that when the training and target data frequency information are similar, the model can achieve good prediction performance. Based on this insight, the paper introduces a novel data synthesis framework to generate training data similar to target data.

**Strengths:**

S1: The paper analyzes the reasons behind the poor zero-shot forecasting performance of existing foundation models and identifies frequency as a factor influencing effective learning from data.

S2: The paper presents a solution to improve zero-shot forecasting performance by synthesizing data based on the specified target frequency, demonstrating good results in experiments.

**Weaknesses:**

W1: The statement 'foundation models often struggle to fully exploit the training distribution, limiting their ability to capture domain-specific patterns' may contradict the fundamental premise of foundation models. Can you provide theoretical support to explain this viewpoint?”

W2: Frequency Generalization: The paper proposes generating synthetic data for training based on a given target frequency, making the synthetic data similar to the target data. This conflicts with the definition of Frequency Generalization.

W3: In Figure 1, neither UniTime nor PatchTST exhibits this phenomenon of Frequency Confusion. Could you explain the reason? Additionally, please clearly describe the experimental details of Figure 1 (right), such as the ratio of target frequency data to W/O target frequency data in the dataset. I believe these experimental settings could significantly impact the results.

W4: Figure 10 visualizes the PCC of the periodogram between datasets. The correlation value between ETTh1 and ETTm1 is roughly between 0.6 and 0.8. Given this, why does the 3rd Choice perform much better than the first choice (0.5-0.9)? Additionally, I don't think the experiments in Figure 2 provide sufficient evidence to conclude that some sector training may help performance.

W5: The basis for generating data is to provide a main target frequency to identify the primary periodicity of the dataset. 1) If the target datasets do not have a dominant frequency and are influenced by multiple frequencies, how should this situation be addressed? 2) If the target dataset lacks periodicity and shows a strong trend instead, how should this situation be addressed? In such cases, this method may be ineffective.

W6: The experiment results: Why do the TTM and Timer results in Table 1 differ significantly from those in the original text under the same experimental setup? Additionally, the results in Table 1 do not outperform some of the latest foundation models, such as Moirai[1], VisionTS[2], and Time-MOE[3].

【1】Unified Training of Universal Time Series Forecasting Transformers

【2】VisionTS: Visual Masked Autoencoders Are Free-Lunch Zero-Shot Time Series Forecasters

【3】Time-MoE: Billion-Scale Time Series Foundation Models with Mixture of Experts

**Questions:**

See W1-W6.

---

### Official Review · Reviewer_sSCq · 2024-11-04

**Soundness:** 2
**Presentation:** 2
**Contribution:** 2
**Rating:** 5
**Confidence:** 4

**Summary:**

This paper studies the factors that influence the learning efficiency of zero-shot time series forecasting. Effective learning includes two perspectives: data and computational efficiency. To answer the question, the authors propose using Fourier analysis to investigate how models learn from synthetic and real-world time series. Through empirical experiments, this paper comes to the conclusion that frequencies in the training data are an important factor that influences performance. Then it proposes to generate data that covers the target frequencies based on its sampling rate. The experiments show the effectiveness of the proposed data generator.

**Strengths:**

s1. The studied problem in this paper is important and practical, since with the development of TSFM, it is important to ask for the generalizability and its influence factors of different models.

s2. It is valuable to see the empirical finding that “forecasters commonly suffer from poor learning from data with multiple frequencies and poor generalization to unseen frequencies”.

s3. Based on its findings, the authors propose a method to generate synthetic data that covers similar frequency information in the target dataset, to improve the overall performance.

**Weaknesses:**

w1. Although the paper comes out with the findings that “forecasters commonly suffer from poor learning from data with multiple frequencies .. ” by experiments, it would be better to add some theoretical analysis to explain why this would happen.

w2. For the results of Figure 2, why do you train GPT4TS, TTM, and UniTime only on one dataset to test the transfer learning performance? While they are TF models, they are supposed to be trained on different datasets. It would be better to expand the experiment to include training on multiple datasets.

w3. Figure 2 is not very intuitive to show the transfer learning ability under different settings.

**Questions:**

Please see w1-w3 in the weakness above.

Another question: based on the findings, does it mean the training dataset should include as many frequencies as possible? If so, it would be better to discuss potential trade-offs or limitations of including many frequencies in the training data, such as increased computational complexity or potential influence.

---

### Official Review · Reviewer_kfCm · 2024-11-06

**Soundness:** 2
**Presentation:** 3
**Contribution:** 3
**Rating:** 5
**Confidence:** 4

**Summary:**

This paper proposes using Fourier analysis to investigate how time series forecasting models learn from synthetic and real-world time series data. The analysis reveals that forecasters commonly suffer from poor learning from data with multiple frequencies (frequency confusion) and poor generalization to unseen frequencies (frequency generalization), which impedes their predictive performance. To alleviate these issues, this paper presents a framework (Freq-Synth) for synthetic data generation, designed to enhance real data or replace it completely by creating task-specific frequency information, requiring only the sampling rate of the target data. Freq-Synth improves the performance of both foundation and non-foundation models in zero-shot and few-shot forecasting.

**Strengths:**

* This paper is well-written. The motivation is clearly described and the proposed analysis and method are easy to follow.

* This paper delves into an important question of ‘What factors govern effective learning in zero-shot time series forecasting?’ It investigates from the perspective of frequencies and gets some findings that may inspire future works.

* The proposed method with synthetic data works well in zero-shot and few-shot settings, which outperforms using real-world data.

**Weaknesses:**

* This paper focuses mostly on few-shot and zero-shot settings, but the term ‘few-shot’ is not clearly defined. How does the analysis or the proposed method perform as the size of training data becomes larger? Will the finding and the improvement be consistent regardless of data size or will there be certain turning points? There need to be more explanations and experiments regarding these questions.

* There is a concern about whether the finding in this paper is general for all different time series data. It may be possible that the evaluated datasets happen to have simple frequency distributions (for example, only very few frequencies are dominant), and thus it is easy to understand that focusing on certain frequencies helps fit certain datasets in such cases. However, would this still be true for datasets with more complex frequency distributions? It would help to evaluate some datasets with more complex frequency distributions.

* The proposed Freq-Synth method still has some limitations. (1) It still needs to train a specific model tailored to a specific dataset, which is different from foundation models that can be applied to various datasets with the same model. (2) It needs some prior knowledge about the target dataset such as the sampling rate or frequencies, which may not always be available. (3) It derives frequencies from the sampling rates with some heuristic ways, which may not always be correct. It is unknown how the method would be influenced if frequencies are estimated incorrectly.

* Some more explanations and experiments are needed for the experimental results. (1) Why do we re-train these foundation models with a subset of real datasets instead of using their original pretraining datasets or using their provided checkpoints? (2) What is the performance if there are more target training data (for example, using all training data in the target dataset for fine-tuning)? (3) Are the results of Freq-Synth still better than those zero-shot or fine-tuning results reported in the original papers of these foundation models? (4) From Tables 1 and 3, it seems that different models do not have clear differences when using Freq-Synth. (5) The proposed method includes some hyper-parameters as in Table 4. It would be better to show how they would influence the performance.

**Questions:**

Please refer to the Weaknesses above.

---

### Author Response · Authors · 2024-11-15

Dear Reviewers,

Thank you for taking the time and effort to review our paper and provide your valuable feedback. We greatly appreciate your insights and constructive comments. After careful consideration, we have decided to withdraw our submission to focus on preparing a major revision that addresses the feedback in depth. We plan to resubmit a significantly improved version in the near future.

Thank you again for your thoughtful reviews and support.

Best regards,
The Authors

---

### Note · Authors · 2024-11-15

I have read and agree with the venue's withdrawal policy on behalf of myself and my co-authors.